# A Design Method for Semi-Rigid Steel Frame via Pre-Established Performance-Based Connection Database

Tulong Yin [1] , Zhan Wang [1], Jianrong Pan [1,*], Kaixiang Zheng [2], Deming Liu [1] and Shengcan Lu [1,3]

1   State Key Laboratory of Subtropical Building Science, South China University of Technology, Guangzhou 510640, China
2   Guangzhou Design Institute Group Co., Ltd., Guangzhou 510620, China
3   School of Civil Engineering and Architecture, Wuyi University, Jiangmen 529020, China
*   Correspondence: ctjrpan@scut.edu.cn

**Abstract:** With decades of research, semi-rigid beam-to-column connections have been widely accepted. However, most studies have been restricted to the local connection level, leaving system-oriented analysis and design methods with a meager investigation, which leads to the fact that the active use of semi-rigid connections in practice is rare. This study aims to provide a system-level design method to bridge the gap between element and connection design, and the two main contributions are to propose a method for designing semi-rigid steel frames by pre-establishing a performance-based connection database and to formulate refined classification criteria for connection performance levels. In this method, the frame design is transformed into finding an appropriate matching of performance requirements between elements and connections. The classification criteria for connection performance levels are based on the assumption that the structural responses (stability, resistance, and deformation) are only slightly affected by the properties of connections within the same level. The emphasis is on the rotational stiffness and moment resistance of the connection. Finally, the results of examples indicate that the connection database is portable and can be applied to various frames, avoiding the repetitive design for connections in different projects. In addition, tuning the performance requirements of the connection can greatly reduce the number of design variables compared to tuning its geometry, and more importantly, it provides designers with a clearer update path, which can significantly shorten the process of trial-and-error and quickly arrive at the final design.

**Keywords:** steel frame design; semi-rigid joint; joint classification criteria; connection database





## 1. Introduction

The beam-to-column joints play an important role in moment resistance frames, not only due to their ability to provide lateral bracing but also their significant influence on the global and local behaviors of the structure in terms of the internal distributions of forces, deformation, and stability [1]. Thus, the simple binary assumption that was used to treat the joints as either perfectly flexible or fully rigid in the past would lead to potential risks in structural design [2,3]. This problem is exacerbated by the widespread use of bolted connections in modern steel structures, as most of them exhibit apparent semi-rigidity. In this context, the current provisions for structural steel buildings worldwide [4–6] clearly state that the real mechanical behavior of the joints should be taken into account in the frame analysis.

Actually, as early as the 1970s, a polynomial model for characterizing the moment-rotation behavior of the connection was proposed by Frye and Morris [7], which is still used by some researchers [8–10]. In the review by Díaz [11], the advantages and disadvantages of various models were summarized, among which the mechanics-based models, such as the Component Method (CM), were considered the most favored by researchers. The CM has been incorporated into the framework of European standards for many years [12],

accompanied by the publication of extensive guidelines [13] and the development of useful computational tables and tools [14,15]. Solving the properties of semi-rigid joints is now a very easy task. Although considerable evolution in the analysis methods of semi-rigid joints has been made, research in this direction is still an open topic, such as the seismic prequalification of dissipative and non-dissipative joints [16], the application of semi-rigid connections in innovative structural forms [17–19], the behavior of semi-rigid joints in spatial steel frames [20,21] or under extreme conditions including fire [22,23] and impact [24,25].

Semi-rigid joints have many advantages in terms of fabrication cost and field erection convenience [26–29]; albeit promising, they lack research on the design methods that can be conveniently applied to the system level, resulting in few reports of their active application in frame design.

Xu et al. [30] firstly used the modified semi-rigid beam element for structural analysis, but his research was more oriented towards analysis rather than design, so the geometry of the joints was not considered. Subsequently, an interactive design method was developed [31]. Within the framework of this approach, designers can iteratively adjust the joint details to change the joint properties, thereby updating the design of frames. Due to a large number of configuration parameters of the joints, their setting has a certain blindness in the preliminary design. To overcome this drawback, Steenhuis et al. [28,32] proposed the concept of joint pre-design. Joint stiffness is first determined by a simplified formula, which is then substituted into the structural analysis. After the element design is completed, further refining for the joints is carried out. On this basis, Bayo et al. [33] relaxed the assumption of joint pre-design to a certain extent and adopted the optimal end fixity-factor of the isolated beam under the action of uniform gravity load as the initial value of the stiffness for joints, which further improved the design efficiency.

In general, all the above methods are the realization of forward thinking; that is, structural analysis is carried out under the given data, including the layout of geometry, the sizes of members, and the details of joints, etc. However, while forward methods seem intuitive and easy to understand, they will encounter a series of difficulties in practical use. A key issue that needs to be addressed in the design of a semi-rigid steel frame is that the elements and connections design must occur simultaneously and iteratively, as the joints fundamentally affect the behavior of the systems. Thus, elements and joints are interdependent; the absence of one makes it impossible to determine the other. Although the pre-design of joints alleviates this dilemma to some extent, the simplified formulas based on additional constraints make it applicable only to the scope that has been used to set up the rules. Another problem is that each joint has many geometric variables, which cannot intuitively reflect their properties, and then inform designers what direction they should adjust next. This blindness will cause designers to get lost in the process of trial-and-error. Therefore, new methods to effectively integrate elements and joint design still need to be developed.

Yin et al. [34] recently proposed an interesting and attractive inverse idea from the perspective of the desired performance of the joints, converting the design of a semi-rigid steel frame into determining the proper match of the elements to the performance requirements of the joints. Since there was no need to consider the configurations of joints explicitly, the complexity of the structural design was significantly reduced. As a continuation of the literature [34], the present study proposes an alternative method for semi-rigid steel frame design via pre-establishing a performance-based connection database, aiming to fill the gap between elements design and semi-rigid joints design and to provide a powerful tool for the promotion of semi-rigid steel frames in practice. The key contribution of this work is that it innovatively proposes a design idea of establishing a standard joint database for given steel sections. This database contains joint details of different performance levels so that in the design process, designers can select joints from the database like members instead of reconstructing joint details in real-time according to the required performance as in literature [34], avoiding the repeated design of joints in different

projects. In addition, through the analysis of the three characteristics of connections in terms of the fixity-factor, flexural capacity coefficient, and ductility, the refined classification criteria for joint performance levels are formulated so that the continuous performance of joints can be approximated by several discrete performance levels, which not only ensures the breadth of the database in terms of performance but also reduces its volume.

The whole process, from the establishment of a connection database to the design of two planar semi-rigid steel frames, is illustrated at the end of the paper. The results are discussed and validated.

## 2. Proposed Method for Semi-Rigid Steel Frame Design

### 2.1. Normalization of the Joint Properties

For a more intuitive understanding of the behavior of semi-rigid joints in structures, two transformations need to be introduced before elaborating on the proposed method for the semi-rigid steel frame design.

Initial rotational stiffness and moment resistance are two important properties of semi-rigid joints, but in the global aspect of the structural system, these two absolute parameters cannot clearly reveal the performance of joints in the structure because the responses of a structure are related to the relative rigidity and flexibility of the internal components that comprise it. Specifically, the rotational behavior of a joint is related to the linear stiffness of the connected beam besides its own rotational stiffness.

Chen [2] firstly adopted the fixity-factor expressed in Equation (1) in the analysis of the semi-rigid steel frame, whose physical meaning is that the ratio of the end rotation of the connected beam, due to an action of unit end moment, to the corresponding rotation of the beam plus the joint.

$$ r = \cfrac{1}{1 + \cfrac{3EI_b}{S_{j,ini}L_b}} \tag{1} $$

where $E$ is Young's modulus; $S_{j,ini}$ is the initial rotational stiffness of the joint obtained by tests, finite element analysis, component method, or any other valid method; $I_b$ is the moment of inertia of the connected beam; $L_b$ is the length of the connected beam.

It is easy to see from Equation (1) that the value of the fixity-factor ranges from zero to one. Zero represents the pinned joint whose stiffness is zero, one represents the rigid joint whose stiffness is infinity, and the intermediate value represents the semi-rigid joint. The advantage of using the fixity-factor is that any type of joint can be equivalent to a bounded real number between zero and one, while using an unbounded value between zero and infinity is difficult.

On the other hand, the moment capacity coefficient of a joint is defined as Equation (2), which indicates the strong/weak relationship between the joint and the connected beam in moment resistance. When this coefficient is less than one, it means a partial strength joint, otherwise a full-strength joint.

$$ m = \frac{M_{j,rd}}{M_{b,pl}} \tag{2} $$

where $M_{j,rd}$ is the moment capacity of the joint, $M_{b,pl}$ is the plastic moment capacity of the connected beam.

The above two relative values are used hereinafter to replace the concepts of rotational stiffness and moment capacity for a joint, respectively.

### 2.2. Philosophy and Methodology

It can be known from the literature [34] that in the semi-rigid steel frame design, it is a more direct and general way to transform the joint design into an inverse problem and then reconstruct the joint details in real-time subject to the required performance. However, another point worth noting is that for a given beam–column section and joint type, there objectively exists a fixed range of rotational stiffness and moment capacity,

as shown in Figure 1. In other words, despite the differences in design conditions and geometrical arrangements for various projects, the performance range and configurations of available joints assembled from the same beam–column section is predeterminable. Therefore, if a connection database for a given set of beam–column profiles is established in advance, conforming with certain criteria, which can be directly queried the available joints according to the required performance. The efficiency can definitely be improved by avoiding the repeated reconstruction of the joint details in each design.

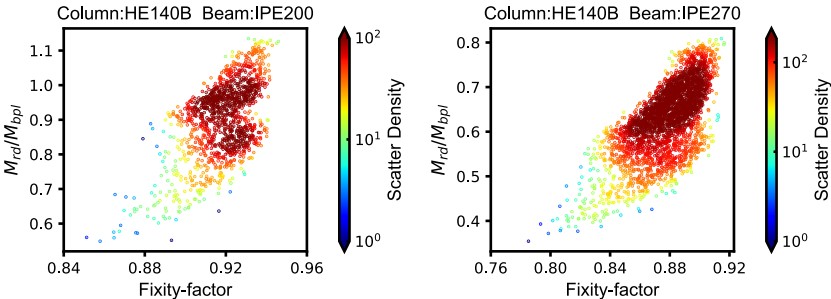

**Figure 1.** Scatter density diagram of fixity-factor vs. moment coefficient of available joints for specific beam–column in Yin et al. [34].

Based on this idea, a design method for a semi-rigid steel frame by querying the connection database is proposed, and its flowchart is shown in Figure 2. Except that task 2 is replaced by querying from the connection database, the rest of the current workflow is the same as in the literature [34]. Therefore, the former inherits all the advantages of the latter. The design of a semi-rigid steel frame is divided into two parts, the element design and the joint design, in sequence. In task 1, joints are abstracted into fixity-factors and their details are implicit; structural analysis can then be carried out. In task 2, the details of each joint can be queried from the pre-generated connection database subject to the required performance provided in the previous stage. The two most critical advantages of this proposed method are: on the one hand, the physical form of a joint is no longer necessary and only manifested as the required performance parameters, which can significantly reduce the number of design variables and make the adjustment direction of the design clearer, resulting in higher solution efficiency. On the other hand, the connection database has good portability and can be flexibly used in various frames, and should be standardized and customized to convert the joint design into the joint query, thereby reducing duplication of work.

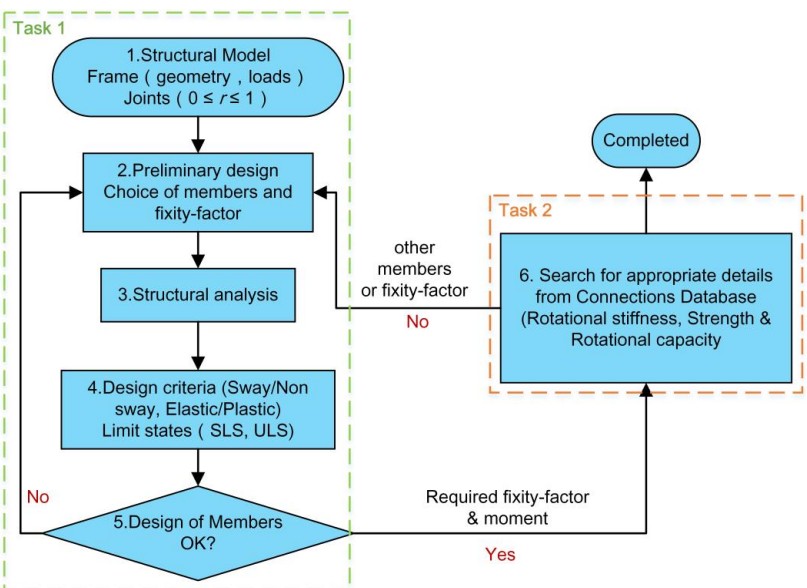

**Figure 2.** Design semi-rigid steel frame via the query of connection database.

## 3. Criteria for Constructing a Performance-Based Connection Database

The three important properties of a joint are initial rotational stiffness, moment resistance, and ductility. Among them, the first two are closely related to the deformation and the internal force distribution of the structure, and the last one is related to the ductility of the structure. EN 1993-1-8:2005 [12] generally classifies joints as rigid, semi-rigid, or pinned in the aspect of stiffness and partial or full strength in the aspect of strength. This classification for properties is beneficial for designers to hold the correct structural concept, but it does not provide more detailed guidance for daily practice because the structural analysis must be based on specific numerical calculations. Therefore, this section will further refine the classification, determine the values of different performance segments, and finally establish the classification criteria that can provide the specific value of performance level. The purpose of dividing the performance level is to approximate a continuous full performance range with a small number of discrete segments, and its core is to determine the interval between segments.

### 3.1. Performance-Levels for Fixity-Factor

The value of the fixity-factor can theoretically be any real number on a continuous domain between zero and one. In practice, however, the properties of available joints should correspond to the discrete points displayed in Figure 1. These points can be clustered according to certain criteria, which is supposed that the structural response is affected only slightly by the stiffness of the joints within the same category.

Steenhuis et al. [32] proposed a "5% resistance criterion" to check whether the stiffness deviation between assumed and 'actual' semi-rigid joints has a significant influence on the frame behavior, whose main process is shown in Figure 3. From the perspective of active control, this means that the deviation of frame bearing capacity can be controlled by limiting the value range of fixity-factor for the 'actual' joints so that all joints in this range can be approximately considered to have the same impact on the structure. Based on this, a single storey single bay frame with pinned supports shown in Figure 4 is firstly considered to illustrate how the variation of joint fixity-factor affects the bearing capacity and deformation of the structure and to deduce the classification criteria for the performance levels of the joint in term of the fixity-factor. Subsequently, the applicability of this result in other support constraints and multi-story multi-bay frames is verified.

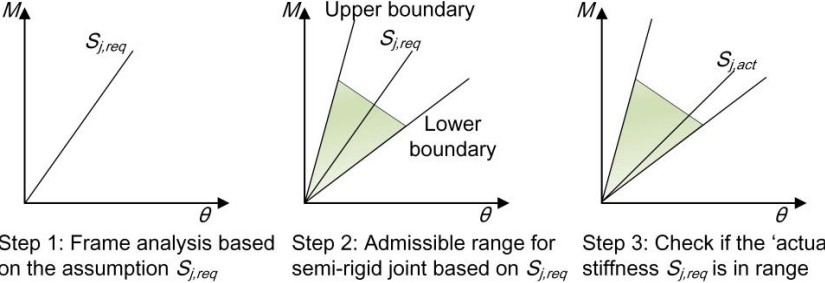

**Figure 3.** Check of a 'actual' semi-rigid joint in Steenhuis et al. [32].

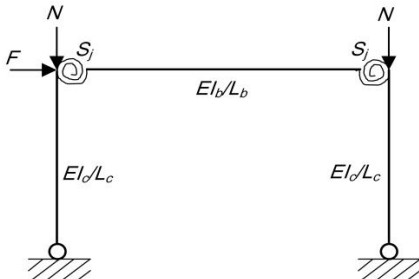

**Figure 4.** Simple frame with semi-rigid beam-to-column joints.

All columns are assumed to share the same profile with a gravity-concentrated load applied on their tops. The rotational stiffness of the beam–column joints is $S_j$ and the line stiffness of beam and column are $k_b = EI_b/L_b$ and $k_c = EI_c/L_c$, respectively. The linear stiffness ratio $\rho$ of beam to column is defined as Equation (3):

$$\rho = \frac{EI_b/L_b}{EI_c/L_c} \tag{3}$$

3.1.1. The Influence of Fixity-Factor Deviation on Column Bearing Capacity

According to reference [35], for a sway frame, the buckling length of the column can be calculated by the following formula:

$$K = \frac{L_{cr}}{L} = \sqrt{\frac{1 - 0.2(\eta_1 + \eta_2) - 0.12\eta_1\eta_2}{1 - 0.8(\eta_1 + \eta_2) + 0.6\eta_1\eta_2}} \tag{4}$$

where $L$ and $L_{cr}$ are the unsupported geometric length and buckling length of the column, respectively; $\eta_1$ and $\eta_2$ are distribution factors. As the supports are pinned, so

$$\eta_1 = 1.0, \ \eta_2 = \frac{k_c}{k_c + k_b} \tag{5}$$

The above formula for calculating $\eta_2$ is based on the assumption that all beam-to-column joints are rigid. For a semi-rigid steel frame, $k_b$ should be multiplied by the following factor [36]:

$$\frac{1.5}{1 + \frac{6EI_b}{L_b S_j}} \tag{6}$$

By substituting Equations (1) and (6) into (5), a new form is obtained:

$$\eta_2 = \frac{4 - 2r}{4 - 2r + 3r\rho} \tag{7}$$

It can be seen from Equation (7) that $\eta_2$ is only related to the fixity-factor and the linear stiffness ratio of beam to column. Moreover, the relationship between the buckling length, the fixity-factor and the linear stiffness ratio can be obtained by substituting Equation (7) into Equation (4).

By using Merchant–Rankine [37] formula, the ultimate resistance of a column can be expressed as:

$$\frac{1}{N_u} = \frac{1}{N_{cr}} + \frac{1}{N_p} \tag{8}$$

where $N_{cr}$ and $N_p$ are the critical elastic buckling load and the squash load of the column, respectively.

Equation (8) can be easily transformed into the following form:

$$N_u = \frac{X}{X + 1}N_p \tag{9}$$

where $X$ is the ratio of critical elastic buckling load to squash load of the column.

The ultimate bearing capacity of a column with joint required or 'actual' fixity-factor can be expressed as Equation (10):

$$N_{u,req} = \frac{X_{req}}{X_{req} + 1}N_p, \ N_{u,act} = \frac{X_{act}}{X_{act} + 1}N_p \tag{10}$$

At the same time, it is easy to obtain the critical elastic buckling load ratio of a column with joint 'actual' or required fixity-factor is defined as ε.

$$\varepsilon = \frac{N_{cr,act}}{N_{cr,req}} = \left(\frac{K_{req}}{K_{act}}\right)^2 \tag{11}$$

where $K_{req}$ and $K_{act}$ are the buckling length coefficient of a column with the required and 'actual' fixity-factor, respectively.

Then, the relative variation of the ultimate bearing capacity of the column under the 'actual' and the required fixity-factor can be expressed as Equation (12):

$$\Delta = \frac{N_{u,act}}{N_{u,req}} - 1 = \frac{\varepsilon - 1}{\varepsilon X_{req} + 1} \tag{12}$$

From Equation (12), it can be found that the variation is related to the fixity-factor and its deviation, the linear stiffness ratio of beam to column, and the ratio of the critical elastic buckling load to the squash load of the column. In practice, the typical ratio between $N_{cr}$ and $N_p$ is in the range of 3 to 17 [38], is 1 even for the very slender column. In addition, the literature [39] suggested that the value $\rho = 0.1$ could be used as the boundary due to the frames for which it holds that $\rho < 0.1$ are not very realistic. Thus, the drop percentages of column resistance corresponding to different fixity-factor under the negative deviations of the two fixity-factors (0.05 and 0.1) are shown in Figure 5.

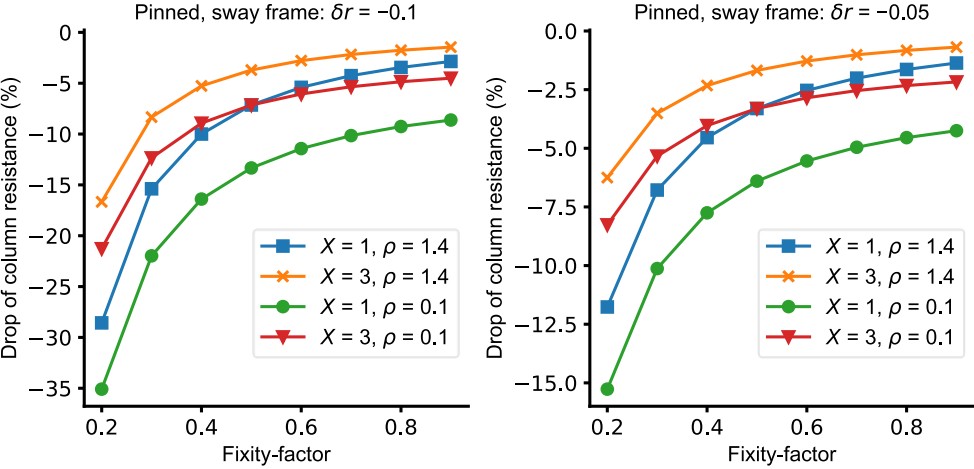

**Figure 5.** Decrease in column resistance due to the negative deviation of the fixity-factor.

As can be seen from Figure 5, a smaller linear stiffness ratio of beam to column results in a larger drop in the column resistance, also for *X*. When the negative deviation between 'actual' and required fixity-factor is 0.05, even for relative slender frames with *X* = 1, $\rho = 0.1$, the column bearing capacity drop can still be controlled within 5.54%, provided that the fixity-factor of the joint is not less than 0.6. On the other hand, when the negative deviation takes the suggested value of 0.1 in Steenhuis et al. [32]. Only if $\rho \geq 1.4$ and joints with a fixity-factor not less than 0.6 are used the column bearing capacity drop can be controlled within 5.41%. While for a relatively slender frame with *X* = 1, $\rho = 0.1$, the drop is obvious, and the minimum drop can be up to 8.62%, which greatly exceeds the "5% resistance criterion".

### 3.1.2. The Influence of Fixity-Factor Deviation on Column Lateral Displacement

For sway frames, in addition to the bearing capacity of the column, its lateral displacement is also sensitive to the change in the stiffness of the beam–column joint. In addition, Jaspart et al. [40] proposed the "10% displacement criterion", which aims to

limit the increase in lateral displacement between the actual frame and the frame with the assumption of the rigid joint within 10%.

As symmetry, the frame in Figure 4 can be equivalent to a column with a rotational spring acting at its top, as shown in Figure 6. The stiffness of the spring shall simultaneously consider the rotational stiffness of the joint and connected beam, which is defined in Equation (13).

$$\frac{1}{S} = \frac{1}{S_j} + \frac{1}{S_{beam}} \tag{13}$$

where $S_{beam}$ is the rotational stiffness of the connected beam, equal to $\frac{6EI_b}{L_b}$.

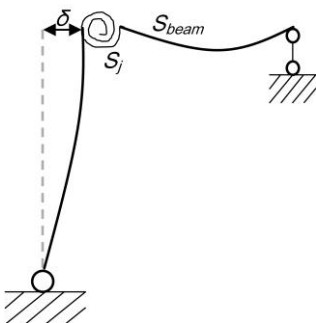

**Figure 6.** An equivalent column with a rotational spring.

The linear elastic analysis is carried out, and the lateral displacement of the column top can be obtained as follows:

$$\delta = \frac{FL_c^3}{2EI_c} \frac{4 - 2r + 4\rho r}{12\rho r} \tag{14}$$

Then, the relative variation of the lateral displacement under the 'actual' and the required fixity-factor can be defined as:

$$\omega = \frac{\delta_{act}}{\delta_{req}} - 1 \tag{15}$$

Substituting Equation (14) into (15), the following formula can be obtained:

$$\omega = \frac{4(r_{req} - r_{act})}{(4 - 2r_{req} + 4\rho r_{req})r_{act}} \tag{16}$$

It can be found from Equation (16) that the variation of the column lateral displacement is related to the fixity-factor and its deviation and the linear stiffness ratio of beam to column. The increase in column lateral displacement corresponding to different fixity-factor under the negative deviation of two fixity-factors (0.025 and 0.05) is shown in Figure 7.

Comparing Figures 5 and 7, when the negative deviation of the fixity-factor is 0.05, it can be found that if $\rho = 0.1$ and the fixity-factor is 0.8, the increase in the column lateral displacement is greater than the drop of its bearing capacity, which are 9.26% and 4.55% respectively. Similar phenomena can also be found when $\rho$ and fixity-factor take other values. This indicates that the increase in the column lateral displacement is more sensitive to the fixity-factor deviation than the decrease in its bearing capacity. The increase in the column lateral displacement caused by the negative deviation of the fixity-factor decreases with the increase in the linear stiffness ratio of the beam to the column. Thus, the frame with $\rho = 0.1$ has the most unfavorable increase in lateral displacement.

When the negative deviation of the fixity-factor is 0.05, the increase in the lateral displacement for relative slender frames with the beam–column linear stiffness ratio of 0.1 and fixity-factor less than 0.8 has to exceed the 10%, while less than 6% for frames whose

beam–column linear stiffness ratio is not less than 1.4 and the fixity-factor is not less than 0.6. If a tighter negative deviation, such as 0.025, is used, the relative increase in lateral displacement can be significantly reduced for all frames. When the fixity-factor is not less than 0.6, the maximum increase in column lateral displacement is less than 6%, which can meet the requirement of the "10% displacement criterion".

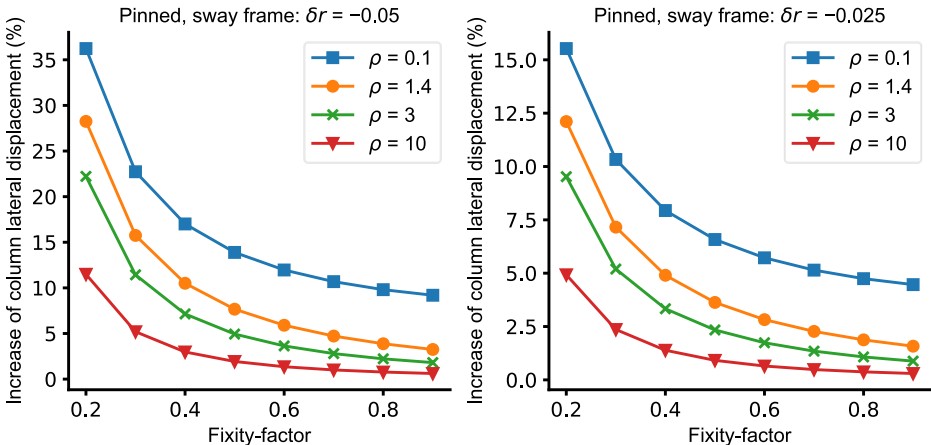

**Figure 7.** Increase of column lateral displacement due to the negative deviation of fixity-factor.

### 3.1.3. The Influence of Bases Constraints

The finding in Sections 3.1.1 and 3.1.2 are all derived from the frame with pin supports. Whether this case would ensure a conservative solution for the frame with fixed support remains to be demonstrated.

Here, the $\eta_1$ in Equation (5) is replaced by zero. Similar to the above, after obtaining the buckling length of the column and substituting it into Equation (12), it is easy to obtain the decrease in the bearing capacity of the column caused by the deviation of the fixity-factor.

The lateral displacement of the column with fixed bases is:

$$\delta_{fixed} = \frac{FL_c^3}{2EI_c} \frac{4 - 2r + 3\rho r}{6(2 - r + 6\rho r)} \tag{17}$$

Substituting Equation (17) into Equation (15), the increase in lateral displacement caused by the deviation of the fixity-factor can be expressed as follows:

$$\omega_{fixed} = \frac{18\rho(r_{req} - r_{act})}{(4 - 2r_{req} + 3\rho r_{req})(2 - r_{act} + 6\rho r_{act})} \tag{18}$$

Figure 8 shows the decrease in column bearing capacity and the increase in column lateral displacement for each fixity-factor under the negative fixity-factor deviation of 0.025. The column bases are fixed. It can be found that the values of the ordinate in the two graphs are much smaller than the results derived from the case of pinned supports, which proves that the inferences in Sections 3.1.1 and 3.1.2 are on the safe side; the tighter deviation limit is not needed.

One thing to note is that the curves with $\rho = 0.1$ in Figure 8, in addition to showing a lower level of variation, also show a different trend from the other types of frames. The smaller the fixity-factor, the less sensitive the column lateral displacement is to its fluctuations. In this case, the linear stiffness ratio of the beam to the column is relatively small, and the rotational constraint of the beam to the column can be almost ignored. Therefore, the column can almost be regarded as a cantilever column with a free upper end and a fixed lower end. It is suggested that the beam–column connections in a sway frame should maintain at least a certain stiffness; otherwise, there may be potential collapse risks for those frames with flexible bases. In this study, the minimum value of the joint fixity-factor is not less than 0.6.

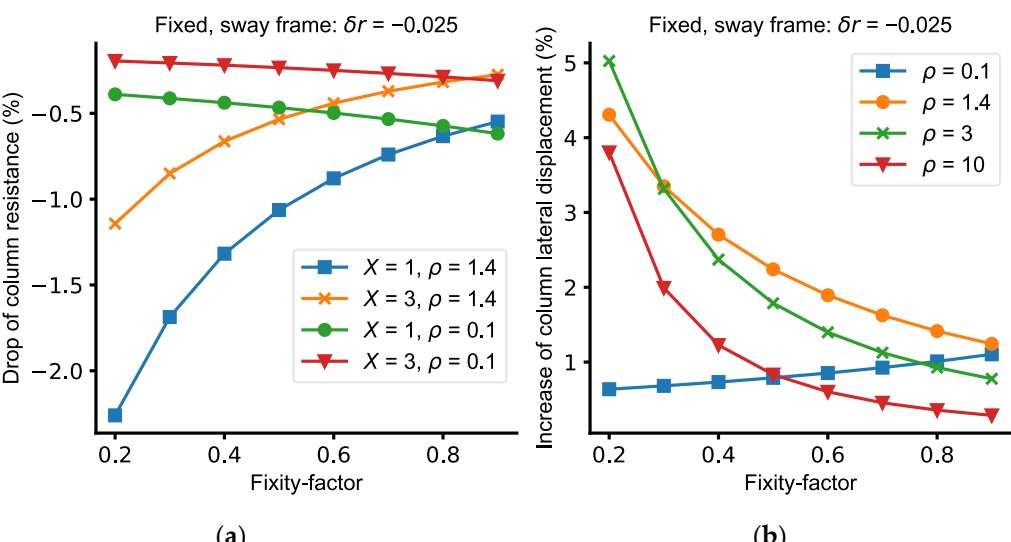

**Figure 8.** Case with fixed base: (**a**) Column resistance dropping, (**b**) Increase of column top lateral displacement.

3.1.4. Applicability Verification in Multi-Story Multi-Bay Frames

In practice, the most common frames are multi-story and multi-bay; thus, different frames need to be analyzed to verify the application of the conclusions drawn from the simple model in complex frames.

From the analysis in Sections 3.1.1–3.1.3, it can be seen that in the simple frame, the column lateral displacement is the most sensitive to the change of the fixity-factor with a low linear stiffness ratio of beam to column and pin bases. Therefore, the frame with $\rho = 0.1$ and pin supports is used as the basis, and the number of stories, spans, and the line stiffness ratio of beam to column is additionally considered to prove whether the variation of the column lateral displacement caused by the negative fixity-factor deviation of 0.025 could meet the requirement of "10% displacement criterion".

Table 1 lists the information on eight structural models. The layout, loading conditions, and labels for elements grouping are all shown in Figure 9. *n* and *m* are the variables number of bays and storeys, respectively. The columns on every two floors are assumed to share the same section, as do the beams. All beam–column joints take one fixity-factor whose value is not less than 0.6. $V_1$ is the roof load with a value of 24.08 kN/m, $V_2$ is the floor load with a value of 44.68 kN/m, and $W$ is the lateral load with a value of 3.29 kN/m. All loads are applied simultaneously, and the combined coefficient of each load is 1.0. Here, eight models are implemented in the commercial structural analysis software SAP2000 v21 [41]. Elastic analysis with the consideration of second-order effects is used. The semi-rigidity of joints can be achieved through the function of "Assign/Frame/Releases and Partial Fixity" in the program. To ensure accuracy, each member should be divided into at least six elements.

Figure 10 shows the increase in the inter-story lateral shift caused by the negative deviation of fixity-factor for three-bay ten-story moment resistance steel frame under different conditions, including various beam–column line stiffness ratios, bases constraints, and fixity-factor. It can be seen that the increase in lateral displacement is less sensitive to the deviation of the fixity-factor when a larger beam–column line stiffness ratio, higher fixity-factor, or more rigid base constraints are used. Figure 11 shows the increase in inter-story lateral shift caused by the negative deviation of the fixity-factor for frames with different total numbers of stories or spans. With the increase in total stories, the magnitude of the increase in the inter-story lateral shift becomes larger, but with the increase in total spans, the opposite trend appears.

**Table 1.** Information for structural models.

| Items | Frame 1 | Frame 2 | Frame 3 | Frame 4 | Frame5 | Frame 6 | Frame 7 | Frame 8 |
|---|---|---|---|---|---|---|---|---|
| $L_b$ (m) | 6.1 | 6.1 | 3.66 | 6.1 | 6.1 | 6.1 | 6.1 | 6.1 |
| $L_c$ (m) | 3.66 | 3.66 | 3.66 | 3.66 | 3.66 | 3.66 | 3.66 | 3.66 |
| $n$ | 3 | 3 | 3 | 3 | 3 | 3 | 1 | 5 |
| $m$ | 10 | 10 | 10 | 10 | 2 | 6 | 10 | 10 |
| Support | pinned | fixed | pinned | pinned | pinned | pinned | pinned | pinned |
| $\rho$ | 0.1 | 0.1 | 0.1~1.55 | 0.1 | 0.1 | 0.1 | 0.1 | 0.1 |
| $r$ | 0.6 | 0.6 | 0.6 | 0.8 | 0.6 | 0.6 | 0.6 | 0.6 |
| C1 | HE550B | HE550B | HE260B | HE550B | HE550B | HE550B | HE550B | HE550B |
| C2 | HE500B | HE500B | HE260B | HE500B | -[1] | HE500B | HE500B | HE500B |
| C3 | HE450B | HE450B | HE260B | HE450B | - | HE450B | HE450B | HE450B |
| C4 | HE400B | HE400B | HE260B | HE400B | - | - | HE400B | HE400B |
| C5 | HE260B | HE260B | HE260B | HE260B | - | - | HE260B | HE260B |
| B1 | IPE400 | IPE400 | IPE400 | IPE400 | IPE400 | IPE400 | IPE400 | IPE400 |
| B2 | IPE360 | IPE360 | IPE360 | IPE360 | - | IPE360 | IPE360 | IPE360 |
| B3 | IPE330 | IPE330 | IPE330 | IPE330 | - | IPE330 | IPE330 | IPE330 |
| B4 | IPE300 | IPE300 | IPE300 | IPE300 | - | - | IPE300 | IPE300 |
| B5 | IPE220 | IPE220 | IPE220 | IPE220 | - | - | IPE220 | IPE220 |

[1] "-" indicates that this component does not exist.

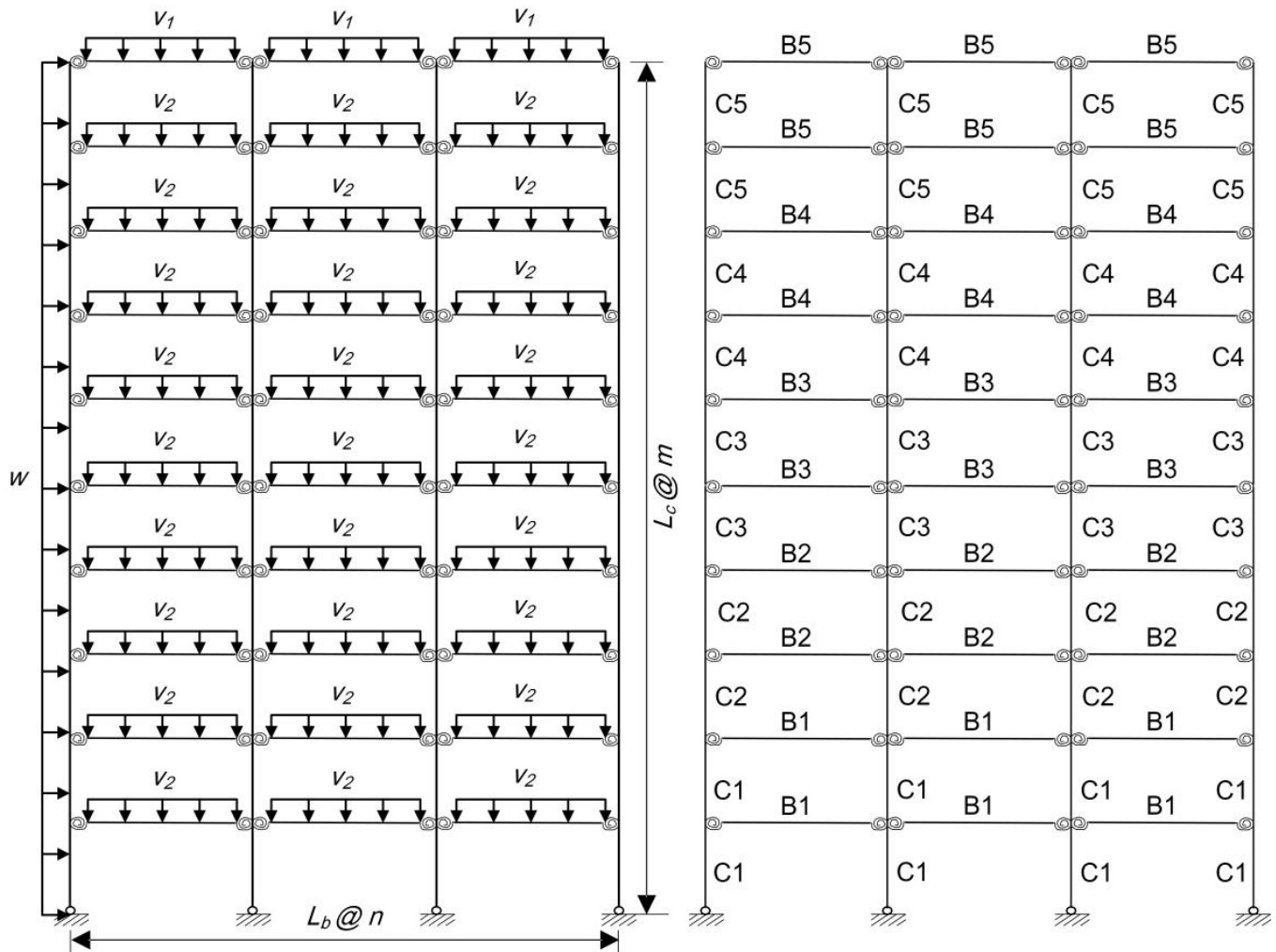

**Figure 9.** The layout of multi-story and multi-bay frame.

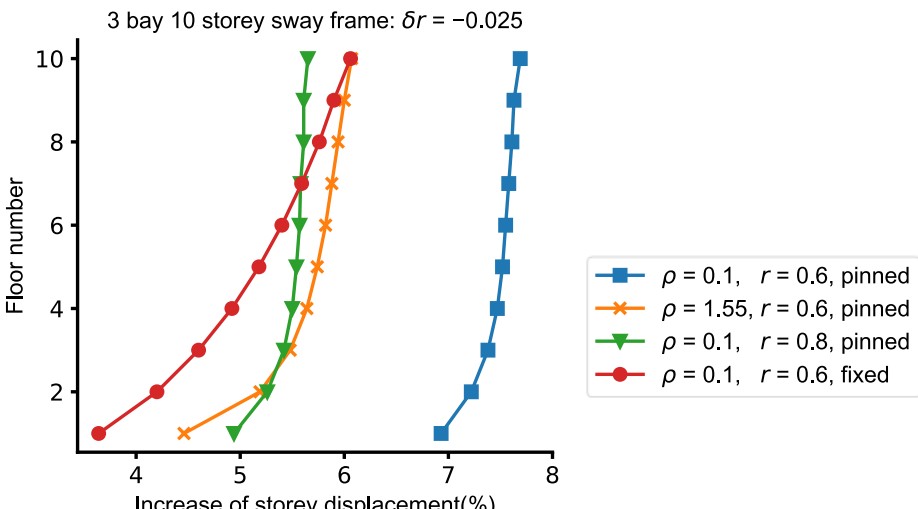

**Figure 10.** Increase of inter-story lateral displacement caused by negative deviation of fixity-factor for the three-span ten-story frame.

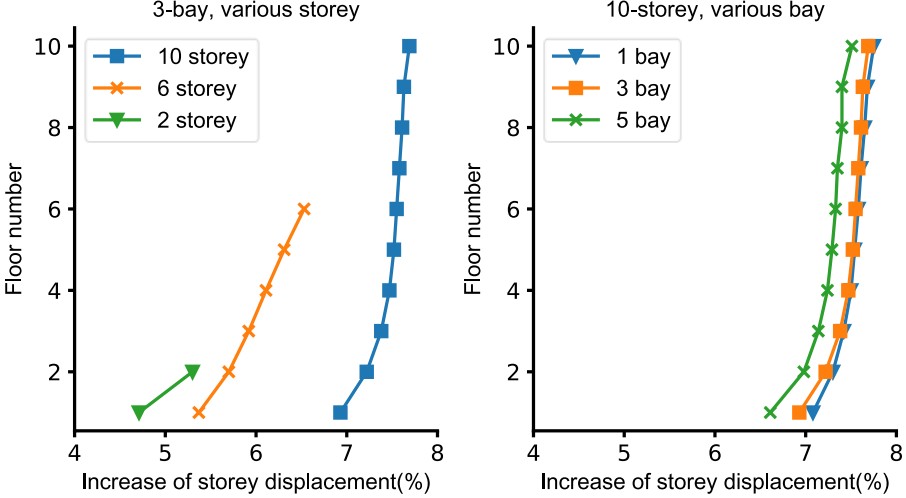

**Figure 11.** An increase in the inter-story lateral shift caused by the negative deviation of the fixity-factor for frames with a different total number of stories or spans. The linear stiffness ratio of the beam to column is 0.1, the fixity-factor is 0.6, and the negative deviation of the fixity-factor is 0.025; the bases are pinned.

Taking all the unfavorable conditions into account, it can be seen that the maximum increase in lateral shift occurs in the single-span ten-story frame, which is 7.75%, but it still meets the requirement of the "10% displacement criterion". This proves that the negative deviation of the fixity-factor derived from the simple frame can be generalized to the multi-span frames below ten stories.

To sum up the above analysis, the principles for dividing performance levels of joint fixity-factor can be determined as follows: (1). To ensure that the frames have a certain capacity of moment resistance, the beam–column joints with low rotational stiffness would not be used. This study recommends that the joint fixity-factor is not less than 0.6. (2). The allowable deviation between the "actual" and required fixity-factor is controlled within ±0.025, which can meet both the "5% resistance criterion" and the "10% displacement criterion".

### 3.2. Performance-Levels for Moment Capacity

The classification for joint moment capacity should follow the hierarchical principle of the resistances of components within a structure. In seismic design, these components are grouped into dissipative (ductile) and non-dissipative (brittle) parts. The core idea

is to dissipate seismic energy through the prior yielding of ductile structural elements to avoid the failure of brittle structural elements. For moment resistance frames, this can be achieved by prioritizing the moment resistance of joints or connected beams. As defined in EN1993:1-8 [12], connections are classified into partial or full strength. When the moment capacity of the connection is less than the plastic moment resistance of the connected beam, it is a partial strength connection; otherwise, it is a full-strength connection.

When designing non-dissipative connections, EN1998-1:2004 indicates [42] that sufficient over-strength is required as expressed in Equation (19):

$$R_d \geq 1.1 \cdot \gamma_{ov} \cdot R_{fy} \tag{19}$$

where $R_d$ is the resistance of the connection; $R_{fy}$ is the plastic resistance of the connected dissipative member; $\gamma_{ov}$ is the over-strength factor; the recommended value is 1.25.

When partial strength connections are used, AISC341-16 [43] points out that in special moment frames, the low limitation for the moment resistance of the connection, determined at the column face, shall not be less than 0.8 times the plastic moment capacity of the connected beam, to avoid severe damage concentration. In the study of Raffaele Landolfo [16], the limit of 0.8 was also adopted, and it was suggested that a lower value of 0.6 could be used at the moment part of the dual frame [44].

For the pursuit of economy, the literature [45] suggested that the end moment requirement of the beam should be as close as possible to the resistance of the connection. At the same time, when the ultimate limit state check is carried out, elastic structural analysis can be performed using the secant stiffness of the connections, which will be explained in Section 4.2. This gives inspiration that the performance levels of joints in terms of moment capacity should ensure that the moment requirement of any joint is greater than 2/3 of its moment capacity.

According to the above discussion, the principles for dividing performance levels of joint moment capacity can be determined as follows: (1). The moment capacity coefficient of the joint is not less than 0.6. (2). The lower level of the joint moment capacity is not less than 2/3 times the higher level.

### 3.3. Ductility Requirement

The ductility requirement of a joint is accompanied by its strength objective. In the traditional seismic design of moment-resistance frames, the dissipation of beam–column joints is not considered, as well as their ductility. If a partial strength joint is used, the joint must have sufficient rotational capacity when the joint is at the plastic hinge location.

EN1993-1-8:2005 [12] suggests a conservative evaluation for the connection with bolted end-plate or angel flange cleat. The ductility of the joint can be ensured by controlling the thickness of either end-plate or column flange or tension flange cleat, provided that the failure of the connection is activated by these components. In a similar way, Mario D'Aniello et al. [46] derived the following formula for ductility evaluation of joints with various moment resistance objectives by taking overstrength and material hardening into consideration:

$$\begin{aligned} t &\leq 0.3d\sqrt{\frac{f_{ub}}{f_y}} \\ t &\leq 0.34d\sqrt{\frac{f_{ub}}{f_y}} \end{aligned} \tag{20}$$

where: $t$ is the thickness of either end-plate or column flange or tension flange cleat; $d$ is the nominal bolt diameter; $f_{ub}$ is the bolt's ultimate strength; $f_y$ is the yield strength of the relevant basic component.

In this study, when generating the connection database, the above formula will be used as the constraint to ensure the ductility requirements of the connections.

### 3.4. Generation of a Connection Database

As shown in Figure 12, the performance levels of the connections include: For fixity-factors are 0.6, 0.65, 0.7, 0.75, 0.8, 0.85, 0.9, and 0.95, which are the midpoint of each level, and the deviation is ±0.025. Note that only half of the first and last segments are taken. Such as 0.7 ± 0.025 represents the segment where the fixity-factor ranges from 0.675 to 0.725, and the influence of all connections within this segment on the structure is approximately equal to that of connections with the fixity-factor of 0.7. For moment capacity coefficients are 0.6, 0.8, 1.0, 1.3, and 1.5, which are the lower bound of each level. Such as, 0.8 represents the connections with a moment capacity coefficient of no less than 0.8 and less than 1.0.

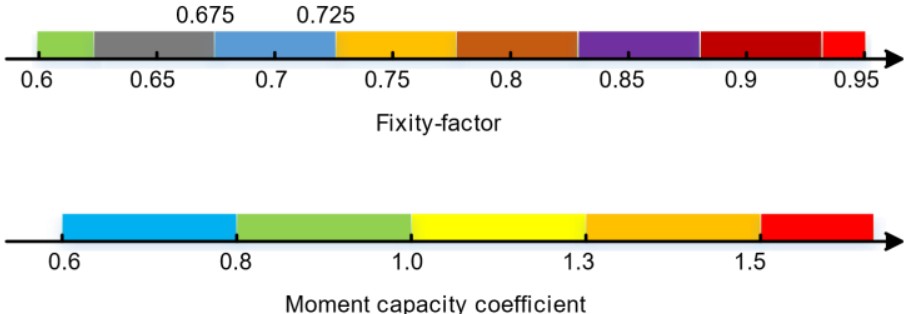

**Figure 12.** Performance segments of connections.

## 4. Joint Modelling

### 4.1. Interaction of Web Panel

As shown in Figure 13, a joint consists of a web panel in shear and a connection. One reasonable joint model should be able to consider the behavior of both fully. In practice, zero-length springs attached to the end of the beams at both sides of the joint are usually used to simulate the rotational characteristics of the connections [36], as shown in Figure 14a. To take into account the deformation of the web panel, in EN1993-1-8:2005 [12], a transformation parameter $\beta$ is defined. However, this parameter is related to the internal force of the structure, and an accurate value must be obtained through iterative calculation. Other models, finite dimensioned four-node joint element [47] and explicit models composed of rigid bars and springs [48–50], are shown in Figure 14b–d. They can naturally consider all forces and deformations that concur at the joint without introducing additional parameters. In this study, the scissor model in Figure 14b is adopted. Since the component of the web is considered separately and relatively simple, the subsequent study of this paper focuses only on the connection part.

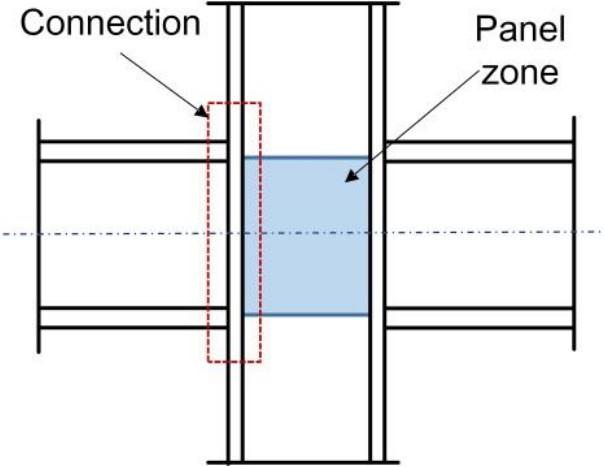

**Figure 13.** Joint (connection + web panel).

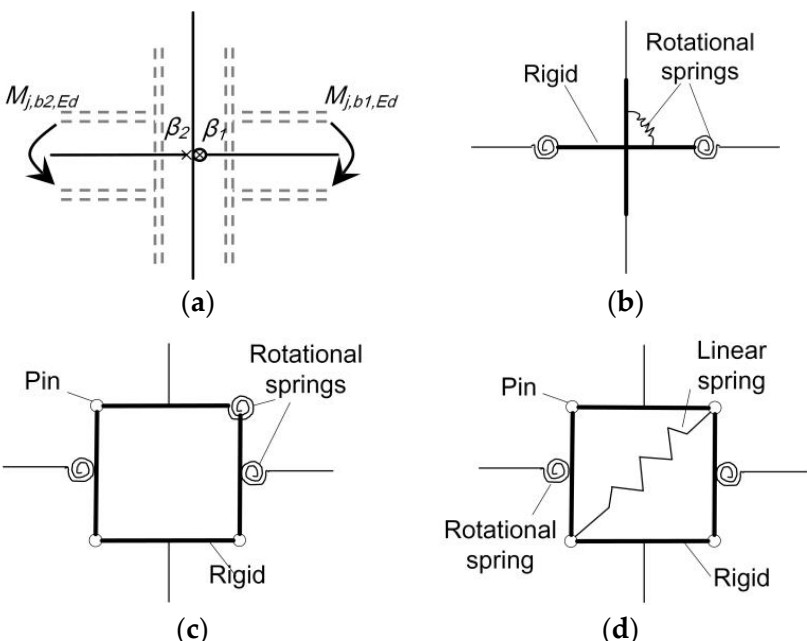

**Figure 14.** Joint modelling: (**a**) Two separate springs; (**b**) Scissor links; (**c**) Rotational spring box; (**d**) Linear spring.

### 4.2. Constitutive Relationship of Rotational Spring

Due to the high nonlinearity of the moment-rotation curve for the connection, the rotational stiffness $S_j$ corresponds to the required bending moment $M_j$ should generally be used in the structural analysis. The most accurate method is, of course, to use the full moment-rotation curve directly, e.g., polynomial or power models were used in some literatures [31,36]. However, these empirical models are only applicable to specific types of connections and require more parameters, which limits the application scope. In this study, a simplified bilinear model, also named the half initial secant method [12], as shown in Figure 15, is adopted. Van Keulen et al. [51] had proved that this model could achieve an appropriate balance between rigor and ease of implementation. The secant stiffness is defined as follows:

$$S_{j,sec} = \frac{S_{j,ini}}{\eta} \tag{21}$$

where: $S_{j,ini}$ is the initial rotational stiffness of the joint; $\eta$ is the reduction factor, which is 2.0 when the required moment is not less than 2/3 joint resistance, otherwise is 1.0.

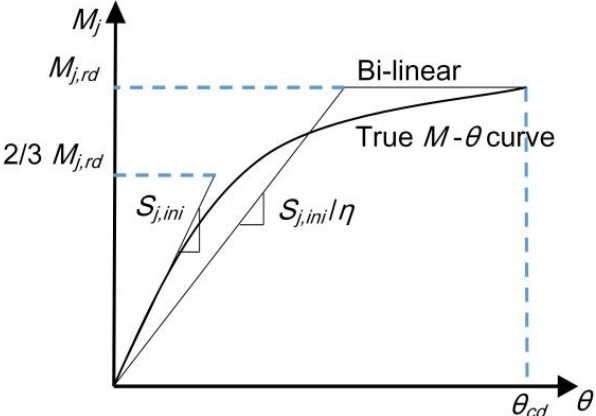

**Figure 15.** Moment-rotation curve of the semi-rigid joint in EN1993-1-8:2005 [12].

## 5. Strategies for Preliminary Design

### 5.1. Pre-Sizing for Members

The pre-sizing for members is largely dependent on the experience of engineers, and some useful rules can be recommended. With a certain reserve, the beam section can be first sized according to the deflection and resistance criteria under the action of gravity loading, and then the column section can be sized by checking compression/buckling resistance under the action of gravity loading with assumed Euler slenderness (60~80 is recommended).

### 5.2. Initial Selection for Fixity-Factor

Yin et al. [34] suggested that for a particular beam–column assembly, selecting the intermediate value of the range in terms of stiffness as the initial fixity-factor can improve the possibility of a successful search for available connections. In particular, for extended end-plate connections, by means of a curve fitting approach to the data, a formula for estimating the initial fixity-factor can be expressed as follow:

$$\ln\left(\frac{1}{r} - 1\right) = A{\cdot}ln\left(b_b t_{bf}\right) + B{\cdot}ln\left(t_{cf}\right) + C - \ln(L_b) \tag{22}$$

where $b_b$ (Unit: mm) is the width of the beam flange; $t_{bf}$ (Unit: mm) is the thickness of beam flange; $t_{cf}$ (Unit: mm) is the thickness of column flange; $L_b$ (Unit: mm) is the length of the connected beam. $A = 0.9875$; $B = 0.1532$; $C = -0.5939$.

Considering that the output of the above formula is a continuous value between zero and one, it can be approximated to the nearest discrete value when combined with the discussion of connection performance levels in Section 3.

## 6. Examples

In this section, two frames are designed to verify the effectiveness of the proposed method, one is from Bayo et al. [33] and the other is from Xu et al. [30]. The joint design conforms to EN1993-1-8:2005 [12], and the steel design conforms to EN1993-1-1:2005 [4]. European sections [52] (i.e., HEB and IPE sections) are used for columns and beams. The inter-story displacement, beam deflection, slenderness, strength, and stability of elements are all checked. Second-order analysis with a buckling length factor of one is adopted, the local bow imperfection of the members is ignored, and the initial swaying imperfection for global analysis of the frame is equivalent to notional horizontal forces. As in Section 3.1.4, two models are implemented in the program SAP2000 v21 [41]. Since the beam-to-column joint is simulated with the scissor model, the properties of the connection and the panel are set by the functions of "Assign/Frame/Releases and Partial Fixity" and "Joint/Panel zones" in the program. The Young's modulus of members can be set to $2.1 \times 10^7$ MPa to approximate rigid bars.

The specific implementation of the method includes two parts: the pre-establishment of the connection database and the design of the semi-rigid steel frame. Here, the connection database only needs to be built once and can be shared in different frames later. The detailed steps are as follows:

- Pre-establishment of the connection database

  Step 1: Determine the set of member sections for which the connection database would be generated.
  Step 2: Select the type of connection (i.e., extended end-plate connection) and reinforcement form of a web panel (i.e., continuity plates).
  Step 3: Select a specific section for beam and column, steel grade, and assume the length of the beam.
  Step 4: Generate the performance matrix for fixity-factor and moment capacity coefficient referring to the discrete levels in Figure 12, and infill each grid with available connections.

Step 5: Go back to Step 3 until all sections are iterated.

- Design of the semi-rigid steel frame

  Step 1: Select a preliminary profile for each member, set the initial value of the fixity-factor according to Section 5.2, then carry out the structural analysis and complete the checking of the rest of the parts except the joints, such as the strength and stability of the members, deformations, etc.

  Step 2: Extract the end moment of each beam, then query the available connections in the database according to the required moment and fixity-factor. If available, the strength check of web panels can also be made.

  Step 3: Check whether all results meet the requirements of the specifications; if not, return to Step 1. Adjust the sections or fixity-factors and re-analyze.

For the frame design, there are two tips to mention: The initial value of the fixity-factor should be reset once any member is changed during the iteration. At the early stage of the design, the reinforcement of the web panel can be temporarily ignored and then be considered if the shear demand exceeds its capacity too much.

### 6.1. A Database for Extended End-Plate Connection

Before the frame design, the connection database should be created in advance. As an example, the asymmetric extended end-plate connection shown in Figure 16 is used in this study. The length of the beam is assumed to be 6.0 m to convert the rotational stiffness into a fixity-factor. It should be noted that this assumption will not affect the applicability of the generated database to other beams with different lengths, as the transformation can be performed using Equation (1).

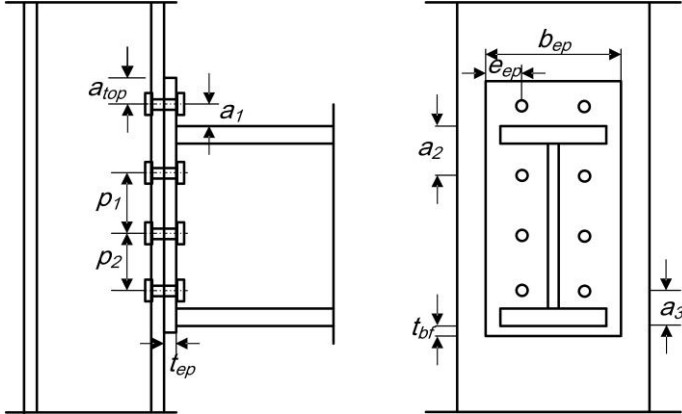

**Figure 16.** The layout of the extended end-plate connection.

Table 2 lists all possible parameter combinations that govern the connection properties. The total row of the bolts on the end plate is not less than three, but the maximum value is not limited, which is related to the beam height. The initial rotational stiffness and the moment resistance of the connection are calculated with reference to EN1993-1-8:2005 [12] and SCI P398 [13]. All welds are at full strength to avoid brittle failure, which is not allowed. To represent an exhaustive solution, the cases of whether to use continuity plates, double plates, and end-plate ribs or not have also been considered. Considering that the reinforcement of the web panel will affect not only the mechanical properties of itself but also the properties of the end connection of the beam, it would be a more comprehensive way to generate a corresponding connection database for different strengthening forms. Currently, only the four forms of webs shown in Figure 17 are considered in this study. For other cases, the corresponding database can also be generated in a similar way. At first glance, the above operations may seem to add complexity to the design, but in reality, the computational cost of generating a connection database is cheap, and more importantly, once completed, it can be used permanently for daily work.

**Table 2.** Parameters for a connection.

| Project | Available |
| --- | --- |
| Plate thickness (mm) | 10, 12, 14, 16, 20, 25 |
| Steel grade [1] | S235, S275, S355 |
| Bolt thread d [2] | M16, M20, M24, M30 |
| Bolt class [3] | 8.8, 10.9 |
| Rows of bolts [4] | $\geq 3$ |
| Edge distance [5] | $1.2d_0 \sim 4t + 40$ |
| Bolt spacing | $\geq 2.2d_0$ |
| Assembly space | $\geq 2.2d_0$ |
| Continuity plates [6] | Yes or No |
| Supplement plates [7] | Single, Double, or No |
| Extended end-plate ribs [8] | Yes or No |

[1] Nominal yield strength of 235, 275, or 355 MPa, only one steel grade per connection. [2] d is the nominal diameter of bolt thread and labeled in accordance with EN 14399-4:2015 [53]. [3] The nominal values of the yield strength and the ultimate tensile strength conform to EN1993-1-8:2005 [12]. [4] There is only one row of bolts at the extended region of the end-plate, the total number of blot rows is not less than three, and the maximum value is related to the connected beam height, not a fixed value. [5] $d_0$ is the diameter of the bolt hole, $t$ is the smaller thickness between the end-plate and the column flange. [6] Minimum value in the list of plate thickness and not less than the thickness of beam flange. [7] Minimum value in the list of plate thickness and not less than the thickness of column web. [8] Minimum value in the list of plate thickness and not less than the thickness of beam web.

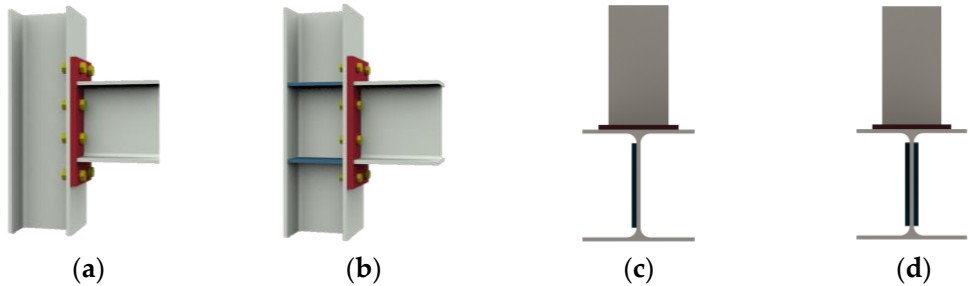

|     |     |     |     |
| :-: | :-: | :-: | :-: |
| (a) | (b) | (c) | (d) |

**Figure 17.** Different forms for web: (**a**) No reinforcement; (**b**) Continuity plates; (**c**) Single supplement plate; (**d**) Double supplement plates.

Through analysis, it can be found that for a particular beam–column assembly, not every performance level has available connections (Two examples can be found in Appendix A). This means that arbitrary performance requirement assumptions can lead to invalid calculations, and finding an appropriate match between the element and the connection performance requirements is the core task of the semi-rigid steel frame design.

*6.2. Three-Bay Two-Storey Frame*

The frame layout, load conditions, and group numbers are shown in Figure 18 [33]. The beams on the same floor share the same profile, and the columns are supposed to be continuous. There are four groups of columns and beams, labeled C1-2 and B1-2, respectively, and eight types of connections and panels, labeled J1-4 and P1-4, respectively. The material is steel S275, with a modulus of elasticity of 210 GPa, and a yield stress of 275 MPa. It is assumed that all beams have sufficient lateral restraint, so their lateral torsional instability is not considered. The maximum allowable ratio of deflection to length for beams is 1/250, the maximum allowable ratio for the inter-story shift to story height is 1/250, and the ratio of roof lateral shift to total height is not more than 1/420. Two load combinations are considered:

$$1.0D_f + 1.0D_r + 0.4L_f + 0.4L_r + 0.5W \text{ for serviceability}$$
$$1.3D_f + 1.3D_r + 1.5L_f + 1.5L_r + 1.5W \text{ for strength}$$

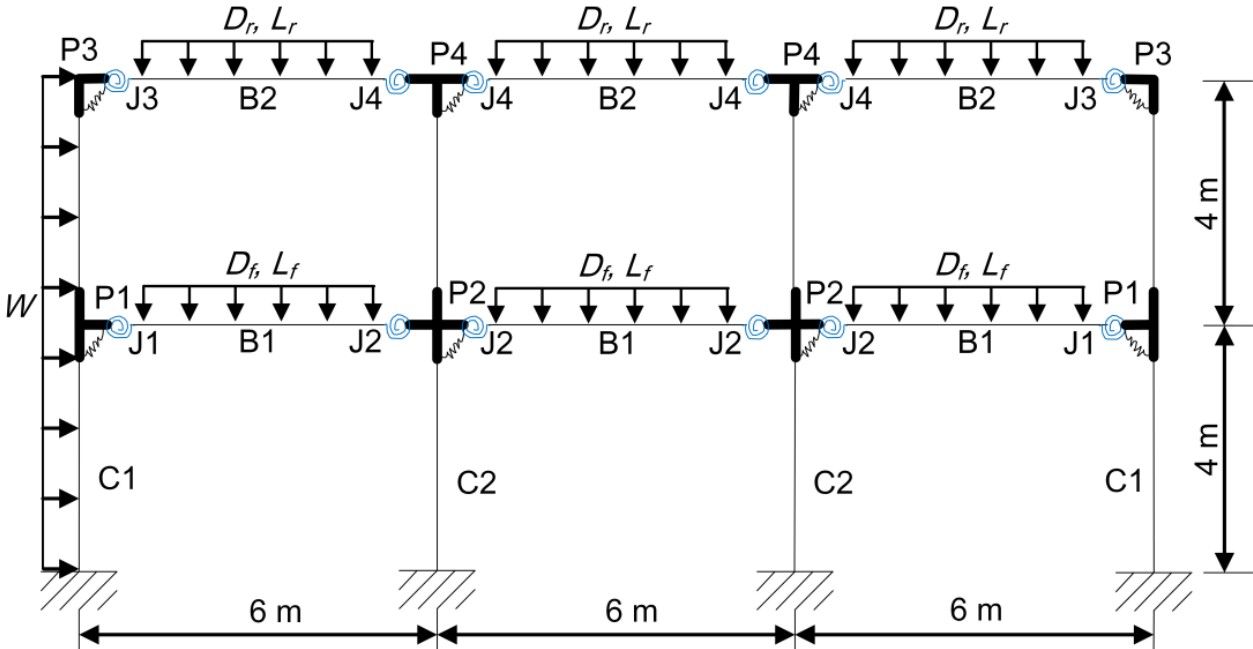

$D_r$ = 6.5 kN/m; $D_f$ = 7.8 kN/m;
$L_r$ = 3.2 kN/m; $L_f$ = 11.2 kN/m;
$W$ = 3.8 kN/m

**Figure 18.** Three-bay, two-story frame.

The information for members, connections, and panels during the iteration is listed in Tables 3–5, respectively. It can be seen from Table 3 that in the initial design, all members have met the required bearing capacity, except that the beams on the second floor are close to the ultimate. However, Tables 4 and 5 show that there is no appropriate solution for the connections on both sides of the internal column on the first floor, and the shear resistance of the panels is also insufficient. On this basis, after tuning the fixity-factor several times, no usable connections were found yet. As a result, the beams on the first floor were replaced with a larger section, as were the inner columns. In Iter1, Table 5 shows that the web panel of the inner column on the first floor is insufficient in shear capacity and needs to be reinforced. After strengthening this position with double plates, a re-analysis was carried out to obtain the final scheme.

Tables 6 and 7 list the final results for the connections and panels, respectively. Since all connections are queried according to the end moment requirements of the beams, the obtained connections are all partial strength. Depending on the ductility requirements of Equation (20), connections with relatively large bolt diameters, classes, and thinner plates are used.

**Table 3.** Members and their most unfavorable ratio of demand to capacity in each iteration.

| No. | Initial | Iter1 | Iter2 |
|---|---|---|---|
| C1 | HEB140 0.67 | HEB140 0.60 | HEB140 0.64 |
| C2 | HEB140 0.80 | HEB160 0.61 | HEB160 0.60 |
| B1 | IPE240 0.87 | IPE270 0.67 | IPE270 0.66 |
| B2 | IPE180 0.96 | IPE200 0.72 | IPE200 0.72 |

**Table 4.** Required and "actual" connection properties in each iteration.

| Phase | No. | $r_{req}$ [1] | $m_{req}$ [2] | $r_{act}$ [3] | $m_{act}$ [4] | Check |
|---|---|---|---|---|---|---|
| Initial | J1 | 0.85 | 0.56 | 0.87 | 0.71 | OK |
| | J2 | 0.85 | 0.83 | Null [5] | Null | NO |
| | J3 | 0.90 | 0.64 | 0.92 | 0.79 | OK |
| | J4 | 0.90 | 0.88 | 0.92 | 0.92 | OK |
| Iter1 | J1 | 0.85 | 0.37 | 0.85 | 0.62 | OK |
| | J2 | 0.85 | 0.64 | 0.87 | 0.74 | OK |
| | J3 | 0.90 | 0.43 | 0.92 | 0.80 | OK |
| | J4 | 0.90 | 0.67 | 0.92 | 0.79 | OK |
| Iter2 | J1 | 0.85 | 0.41 | 0.87 | 0.70 | OK |
| | J2 | 0.85 | 0.63 | 0.87 | 0.74 | OK |
| | J3 | 0.90 | 0.44 | 0.92 | 0.80 | OK |
| | J4 | 0.90 | 0.66 | 0.92 | 0.79 | OK |

[1] Required fixity-factor. [2] Required moment capacity coefficient. [3] 'Actual' fixity-factor. [4] 'Actual' moment capacity coefficient. [5] "Null" indicates that there is no available connection for the given performance requirements.

**Table 5.** Required web panel in each iteration.

| Phase | No. | $M_{rd,req}$ [1] | $M_{rd,act}$ [2] | Continuity [3] | Double [4] | Check |
|---|---|---|---|---|---|---|
| Initial | P1 | 61.89 | Null [5] | Null | Null | NO |
| | P2 | 31.46 | Null | Null | Null | NO |
| | P3 | 31.98 | 33.75 | 0 | 0 | OK |
| | P4 | 10.77 | 70.88 | 1 | 0 | OK |
| Iter1 | P1 | 54.15 | Null | Null | Null | NO |
| | P2 | 34.58 | 68.06 | 0 | 0 | OK |
| | P3 | 28.75 | 37.50 | 0 | 0 | OK |
| | P4 | 11.57 | 50.41 | 0 | 0 | OK |
| Iter2 | P1 | 60.67 | 75.47 | 0 | 1 | OK |
| | P2 | 32.90 | 68.06 | 0 | 0 | OK |
| | P3 | 29.22 | 37.50 | 0 | 0 | OK |
| | P4 | 10.95 | 50.41 | 0 | 0 | OK |

[1] Required moment capacity (kNm). [2] Actual moment capacity (kNm). [3] "0" and "1" respectively indicate the presence or absence of the continuity plates. [4] "0", "1," and "2," respectively indicate none, single and double of the supplement plate. [5] "Null" indicates that there is no available connection for the given performance requirements.

**Table 6.** Final connection details (size in mm).

| Joint | Beam | Column | Bolt [1] | $t_{ep}$ | $b_{ep}$ | $e_{ep}$ | $a_{top}$ | $a_1$ | $a_2$ | $a_3$ |
|---|---|---|---|---|---|---|---|---|---|---|
| J1 | IPE270 | HEB140 | M20 | 10 | 136 | 28 | 65 | 27 | 59 | 50 |
| J2 | IPE270 | HEB160 | M20 | 10 | 155 | 37 | 77 | 27 | 55 | 48 |
| J3 | IPE200 | HEB140 | M20 | 10 | 140 | 35 | 59 | 48 | 42 | 35 |
| J4 | IPE200 | HEB160 | M20 | 10 | 149 | 38 | 28 | 46 | 36 | 44 |

[1] Bolt class is 10.9.

**Table 7.** Final web panel details (size in mm).

| No. | Continuity Plate [1] | Double Plate [2] |
|---|---|---|
| P1 | Null [3] | $1 \times 10$ |
| P2 | Null | Null |
| P3 | Null | Null |
| P4 | Null | Null |

[1] Width $\times$ thickness, the height is equal to the net height of the column web by default. [2] Numbers of block $\times$ thickness, the height of the plate is the same as that of the end-plate, and the width is equal to the net height of the column web by default. [3] "Null" indicates the component is not needed.

Figure 19 shows the relative difference in responses of the frame under the required and the obtained joints' performance, including internal forces, lateral displacements, and the most unfavorable ratios of demand to capacity for members. All values are within 5% and meet the engineering tolerance, proving the effectiveness of the proposed method.

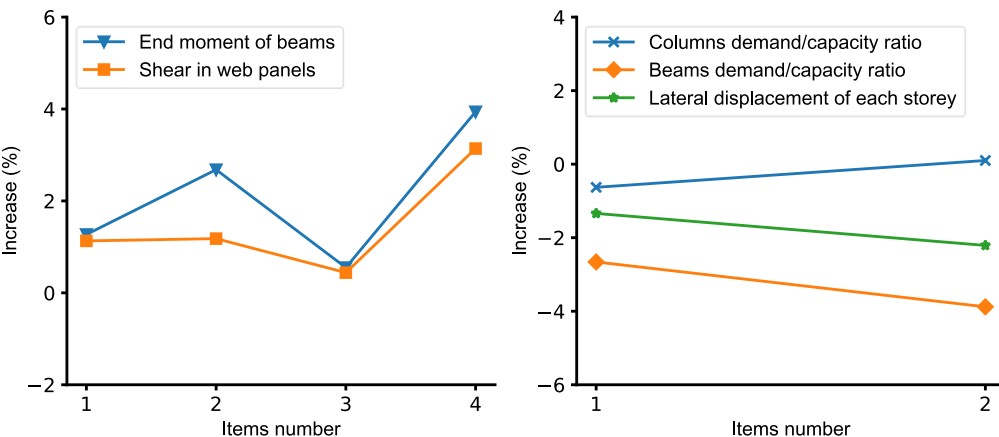

**Figure 19.** The relative difference for responses of the frame under the required and the obtained joints performance.

*6.3. Three-Bay Ten-Story Frame*

This example aims to verify the applicability of the connection database in a high-rise frame with more variables. The frame layout, load conditions, and group numbers are shown in Figure 20 [30]. The beams on the same floor share the same profile and the columns on every two floors are the same but different between side and internal. There are twenty groups of columns and beams, labeled C1-10 and B1-10, respectively, and forty types of connections and panels, labeled J1-20 and P1-20, respectively. The material is steel S235, with a modulus of elasticity of 210 GPa and a yield stress of 235 MPa. It is assumed that all beams have sufficient lateral restraint, so their lateral torsional instability is not considered. The maximum allowable ratio of deflection to length for beams is 1/250, the maximum allowable ratio for inter-story shift to story height is 1/250, and the ratio of roof lateral shift to total height is not more than 1/420. Two load combinations are considered:

$$1.0D_f + 1.0D_r + 0.4L_f + 0.4L_r + 0.5W \text{ for serviceability}$$
$$1.2D_f + 1.2D_r + 0.5L_f + 0.5L_r + 1.3W \text{ for strength}$$

Due to the larger number of members and joints in this example, only the main features of its design process are summarized here, and the detailed iterations can be found in Appendix B. This frame has undergone four major iterations, from the initial design to the final solution. Similar to the previous example in Section 6.2, it can be found that it is relatively easy to find suitable sections for the members, which can usually be made after the initial adjustment. Then, according to the strategies for selecting the initial fixity-factor proposed in Section 5.2, the available connections can also be found quickly. The most complicated part is the panels, which require an iterative trade-off between choosing a larger section or localized reinforcement. Even though the complexity of the design increases significantly due to the increase in the number of members, connections, and panels to be controlled, the proposed method can still quickly arrive at the final design within relatively few iterations. All of these, just by tuning the geometric parameters of the connections, will leave designers lost in the minutiae, conversely, the direction of the design can be made clearer by controlling their performance requirements.

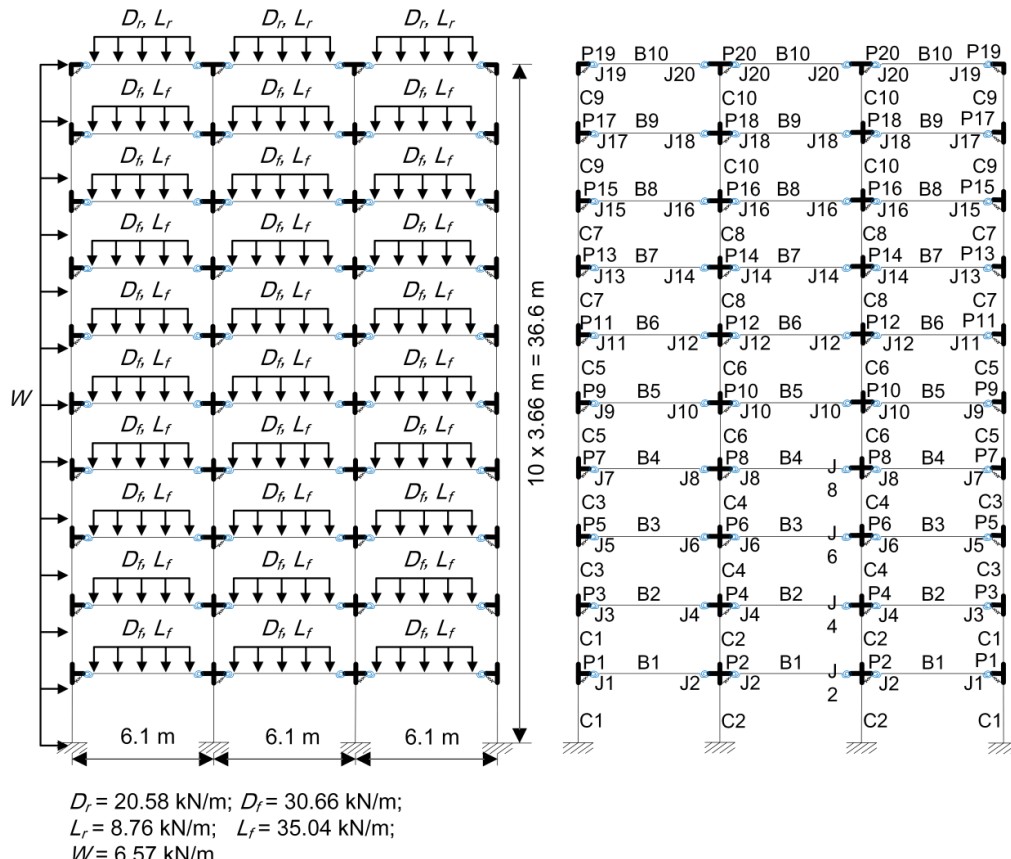

$D_r$ = 20.58 kN/m; $D_f$ = 30.66 kN/m;
$L_r$ = 8.76 kN/m;   $L_f$ = 35.04 kN/m;
$W$ = 6.57 kN/m

**Figure 20.** Three-bay ten-story frame.

Tables 8 and 9 list the final results for the connections and panels, respectively. Similar to Section 6.2, due to the ductility requirements for partial strength connections, relatively large bolt diameters, classes, and thinner end plates are used.

**Table 8.** Final connection details (size in mm).

| Joint | Beam | Column | Bolt [1] | $t_{ep}$ | $b_{ep}$ | $e_{ep}$ | $a_{top}$ | $a_1$ | $a_2$ | $a_3$ | $p_1$ | $p_2$ | $p_3$ |
|---|---|---|---|---|---|---|---|---|---|---|---|---|---|
| J1 | IPE360 | HE300B | M24 | 14 | 255 | 35 | 78 | 51 | 73 | 71 | 109 | 107 | Null [2] |
| J2 | IPE360 | HE900B | M30 | 16 | 270 | 69 | 83 | 76 | 59 | 73 | 142 | 86 | Null |
| J3 | IPE400 | HE300B | M24 | 14 | 295 | 50 | 52 | 47 | 59 | 46 | 86 | 132 | 77 |
| J4 | IPE400 | HE900B | M24 | 14 | 195 | 38 | 57 | 38 | 48 | 69 | 159 | 124 | Null |
| J5 | IPE400 | HE260B | M24 | 14 | 237 | 39 | 90 | 50 | 69 | 50 | 66 | 96 | 119 |
| J6 | IPE400 | HE500B | M24 | 14 | 270 | 41 | 67 | 49 | 49 | 60 | 117 | 89 | 85 |
| J7 | IPE400 | HE260B | M24 | 14 | 237 | 39 | 90 | 50 | 69 | 50 | 66 | 96 | 119 |
| J8 | IPE400 | HE500B | M24 | 14 | 270 | 41 | 67 | 49 | 49 | 60 | 117 | 89 | 85 |
| J9 | IPE400 | HE240B | M30 | 16 | 225 | 42 | 89 | 55 | 89 | 70 | 124 | 117 | Null |
| J10 | IPE400 | HE340B | M30 | 16 | 280 | 44 | 70 | 74 | 68 | 55 | 188 | 89 | Null |
| J11 | IPE400 | HE240B | M24 | 14 | 231 | 38 | 67 | 54 | 67 | 69 | 132 | 132 | Null |
| J12 | IPE400 | HE340B | M30 | 16 | 280 | 44 | 70 | 74 | 68 | 55 | 188 | 89 | Null |
| J13 | IPE360 | HE200B | M24 | 14 | 186 | 34 | 59 | 56 | 73 | 76 | 58 | 153 | Null |
| J14 | IPE360 | HE260B | M30 | 16 | 235 | 41 | 74 | 76 | 78 | 85 | 111 | 86 | Null |
| J15 | IPE360 | HE200B | M24 | 14 | 186 | 34 | 59 | 56 | 73 | 76 | 58 | 153 | Null |
| J16 | IPE360 | HE260B | M30 | 16 | 235 | 41 | 74 | 76 | 78 | 85 | 111 | 86 | Null |
| J17 | IPE360 | HE180B | M24 | 14 | 178 | 32 | 55 | 50 | 74 | 68 | 112 | 106 | Null |
| J18 | IPE360 | HE180B | M24 | 25 | 176 | 32 | 59 | 62 | 67 | 67 | 88 | 138 | Null |
| J19 | IPE270 | HE180B | M20 | 12 | 173 | 30 | 79 | 28 | 62 | 62 | 146 | Null | Null |
| J20 | IPE270 | HE180B | M20 | 12 | 173 | 30 | 79 | 28 | 62 | 62 | 146 | Null | Null |

[1] Bolt class is 10.9. [2] "Null" indicates this parameter is not needed.

**Table 9.** Final web panel details (size in mm).

| Label | Continuity Plate | Double Plate |
|---|---|---|
| P1 | $130 \times 12$ | Null |
| P2 | Null | Null |
| P3 | Null | Null |
| P4 | $132 \times 14$ | Null |
| P5 | Null | $1 \times 10$ |
| P6 | Null | Null |
| P7 | Null | $1 \times 10$ |
| P8 | Null | Null |
| P9 | $110 \times 14$ | Null |
| P10 | Null | Null |
| P11 | Null | $1 \times 10$ |
| P12 | Null | Null |
| P13 | $82 \times 12$ | Null |
| P14 | Null | Null |
| P15 | $82 \times 12$ | Null |
| P16 | Null | Null |
| P17 | Null | $1 \times 10$ |
| P18 | Null | Null |
| P19 | Null | Null |
| P20 | Null | Null |

Figure 21 shows the relative difference in responses of the frame under the required and the obtained joints' performance, including internal forces, lateral displacements, and the most unfavorable ratios of demand to capacity for members. All values are within 5% and meet the engineering tolerance, proving the effectiveness of the proposed method.

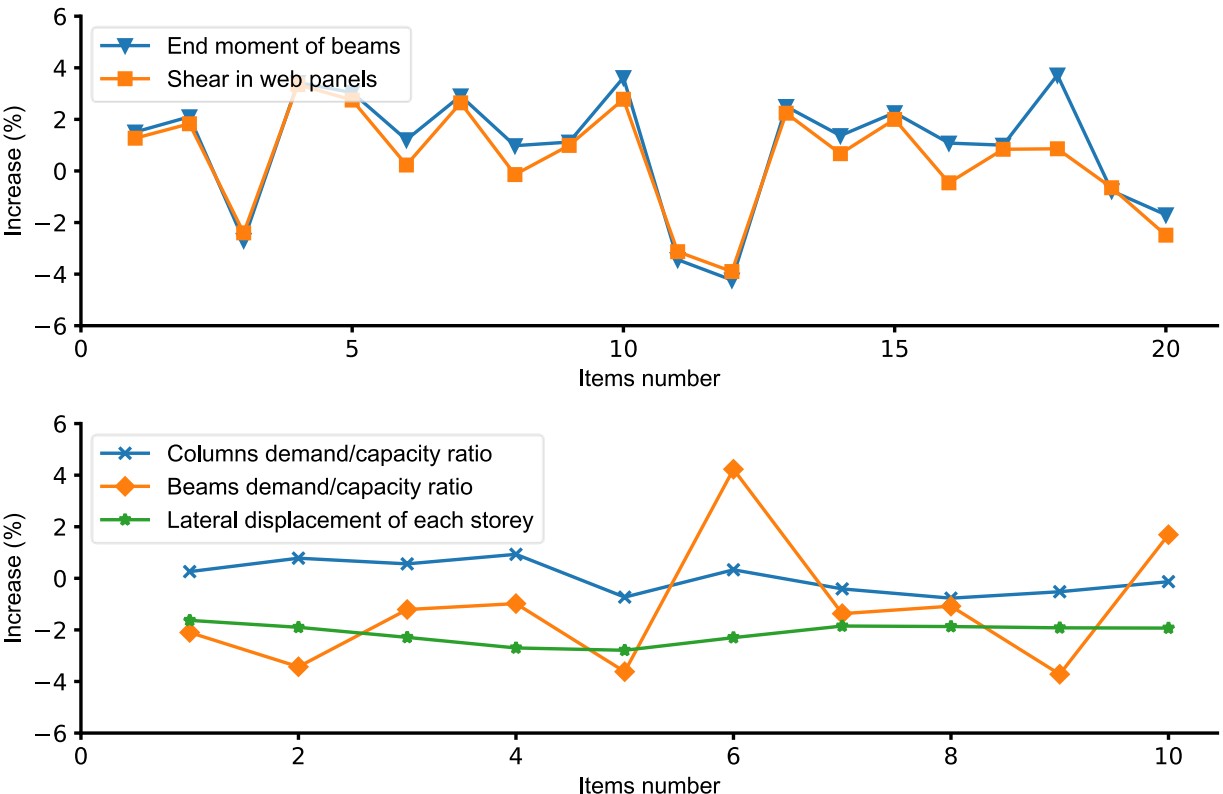

**Figure 21.** The relative difference for responses of the frame under the required and the obtained joints performance.

## 7. Conclusions

This paper presents a design method for semi-rigid steel frames by building a performance-based connection database in advance. Based on the results, the following conclusions can be drawn.

1.  As a continuation of the work in literature [34], the new method proposed in this paper has additional features besides maintaining all the advantages of the old one. Under the framework of the proposed method, the structural design is separated into two consecutive parts, element, and joint design, which are linked together by the performance requirements of joints. The structural design is transformed into a search for the proper matching of performance requirements between elements and joints, while the geometry of connections no longer needs to be concerned throughout the process. As a portability template, once the connection database is generated, it can be easily used for various frames, avoiding the repeated design of connections in different projects.

2.  The general principle for classifying the performance levels of connections is established, whose core is to determine the interval between the upper and lower levels, followed by the applicability boundaries. In terms of the joint fixity-factor, it is found that the lateral displacement of the frame is more sensitive to the variation of the fixity-factor than internal forces. Frames with a low stiffness ratio of the beam to column are more sensitive to the variation of the fixity-factor than those with a high ratio. The sensitivity increases with the total number of floors and, conversely, decreases with the total number of spans. For multi-story multi-bay moment resistance frames with less than ten stories, when the fixity-factor is not less than 0.6, all connections with a difference in the fixity-factor within ±0.025 can be classified into one category. In terms of the moment capacity coefficient, it is stipulated that the lower level should not be less than 2/3 times that of the higher level so that the obtained connection requirements can be as close as possible to its resistance and the flexural capacity of the connection is fully utilized.

3.  The performance matrix for connections is established, whose vertical axis is the fixity-factor with the value of 0.6, 0.65, 0.7, 0.75, 0.8, 0.85, 0.9, and 0.95, and the horizontal axis is the moment capacity coefficient with the value of 0.6, 0.8, 1.0, 1.3, and 1.5. For each assembly of beam–column, not all performance grids have available connections. This means that, on the one hand, the volume of the connection database can be compressed, and on the other, arbitrary assumed performance requirements for the connections can lead to invalid calculations, while choosing a good initial value is tricky.

4.  The design process of two frames demonstrates the feasibility and high efficiency of the proposed method. The design of joints needs more iterations than that for elements to obtain a satisfactory solution. Due to the ductility requirements for partial strength connections, large diameters, high classes of bolts, and thin plates are used. The changes in the structural responses caused by the deviation between the obtained connections and the assumed connections are all within 5%, which can meet the engineering requirements well.

In this study, only the use of an extended end-plate connection is shown. In fact, this method can be easily extended to any type of connection as long as the corresponding database is generated, so that hybrid connections can be used in a frame, making the design more realistic and reasonable. In addition, the examples only show the use of the connection database in the elastic design. When the plastic design is carried out, the fixity-factor and moment capacity coefficient of the connection should be introduced into the structural model simultaneously, and the strategies for tuning these two parameters efficiently still need to be explored.

**Author Contributions:** Conceptualization, T.Y. and Z.W.; Data curation, T.Y.; Methodology, T.Y.; Software, T.Y. and K.Z.; Validation, T.Y. and S.L.; Formal analysis, T.Y.; Investigation, T.Y.; Resources, T.Y. and K.Z.; Writing—original draft preparation, T.Y.; Writing—review and editing, T.Y., D.L., J.P. and S.L.; Visualization, T.Y. and D.L.; Supervision, Z.W.; Project administration, Z.W. and J.P.; Funding acquisition, Z.W. and J.P. All authors have read and agreed to the published version of the manuscript.

**Funding:** This research was funded by the National Natural Science Foundation of China (Grant No. 51978279, 52278181), Guangdong Basic and Applied Basic Research Foundation (Grant No. 2020A1515011307, 2021B1212040003), the State Key Laboratory of Subtropical Building Science (Grant No. 2022KA04, 2022KB15).

**Institutional Review Board Statement:** Not applicable.

**Informed Consent Statement:** Not applicable.

**Data Availability Statement:** The data presented in this study are available on request from the corresponding author. The data are not publicly available due to privacy.

**Acknowledgments:** Not applicable.

**Conflicts of Interest:** The authors declare no conflict of interest.

## Nomenclature

| | | | |
|---|---|---|---|
| $E$ | Young's modulus | $N$ | vertical load on top of the column |
| $r$ | end fixity-factor | $N_u$ | ultimate resistance of the column |
| $\delta r$ | deviation between the required and 'actual' fixity-factor | $N_{u,req}$ | column ultimate resistance at the required fixity-factor |
| | | $N_{u,act}$ | column ultimate resistance at the 'actual' fixity-factor |
| $r_{req}$ | required fixity-factor | $\varepsilon$ | column critical elastic buckling load ratio at the required and 'actual' fixity-factor |
| $r_{act}$ | 'actual' fixity-factor | | |
| $m$ | moment capacity coefficient | $N_{cr}$ | critical elastic buckling load of the column |
| $\eta$ | reduction factor for the joint secant stiffness | $N_p$ | squash load of the column |
| $S_{j,ini}$ | initial rotational stiffness of the joint | $X$ | ratio of the critical elastic buckling load to the squash load |
| $S_{j,sec}$ | secant stiffness of the joint | $X_{req}$ | ratio of the critical elastic buckling load to the squash load at the required fixity-factor |
| $M_{j,rd}$ | moment resistance of the joint | | |
| $M_{b,pl}$ | plastic moment resistance of the beam | $X_{act}$ | ratio of the critical elastic buckling load to the squash load at the 'actual' fixity-factor |
| $k_b$ | linear stiffness of the beam | $\Delta$ | relative variation in the ultimate bearing capacity of the column due to the deviation between the 'actual' and the required fixity-factor |
| $k_c$ | linear stiffness of the column | | |
| $\rho$ | linear stiffness ratio of beam to column | | |
| $I_b$ | moment of inertia of the beam | | |
| $I_c$ | moment of inertia of the column | $\delta$ | lateral displacement of the column |
| $L_b$ | length of the beam | $\delta_{act}$ | column lateral displacement at the 'actual' fixity-factor |
| $L_c$ | length of the column | $\delta_{req}$ | column lateral displacement at the required fixity-factor |
| $L$ | unsupported length of the column | $\omega$ | relative variation in the lateral displacement of the column due to the deviation between the 'actual' and the |
| $L_{cr}$ | Euler buckling length of the column | | |

| | | |
|---|---|---|
| $\eta_1, \eta_2$ | distribution factors of the lower and upper ends of the column | required fixity-factor |
| $K$ | effective length factor | $d$   nominal bolt diameter |
| $K_{req}$ | column effective length factor at the required fixity-factor | $t$   thickness of the end-plate |
| | | $f_{ub}$   bolt ultimate strength |
| | | $f_y$   yield strength of the end-plate or column flange |
| $K_{act}$ | column effective length factor at the 'actual' fixity-factor | $\gamma_{ov}$   over-strength factor |
| | | $R_d$   resistance of the connection |
| $F$ | lateral load on top of the column | $R_{fy}$   plastic resistance of the connected dissipative member |

## Appendix A

Due to the huge volume of the connection database, it is impossible to list all details; only two examples are listed in Tables A1 and A2, respectively. For a particular beam–column assembly, such as the HEB160 column and IPE200 beam, not every performance level has available connections.

**Table A1.** The performance matrix for connections assembled by HEB160 column and IPE200 beam, without any web reinforcement, and steel grade is S275.

| $r\backslash m$ | 0.6 | 0.8 | 1.0 | 1.3 | 1.5 |
|---|---|---|---|---|---|
| 0.6 | | | | | |
| 0.65 | | | | | |
| 0.7 | | | | | |
| 0.75 | | | | | |
| 0.8 | | | | | |
| 0.85 | $\surd$ [1] | | | | |
| 0.9 | $\surd$ | $\surd$ | $\surd$ | | |
| 0.95 | | | $\surd$ | | |

[1] "$\surd$" indicates that there are available connections for this required performance.

**Table A2.** Performance matrix for connections assembled by HEB160 column and IPE200 beam, with continuity plates, and steel grade is S275.

| $r\backslash m$ | 0.6 | 0.8 | 1.0 | 1.3 | 1.5 |
|---|---|---|---|---|---|
| 0.6 | | | | | |
| 0.65 | | | | | |
| 0.7 | | | | | |
| 0.75 | | | | | |
| 0.8 | | | | | |
| 0.85 | | | | | |
| 0.9 | $\surd$ | $\surd$ | $\surd$ | | |
| 0.95 | $\surd$ | $\surd$ | $\surd$ | | |

## Appendix B

The detailed design process of the example in Section 6.3 is shown as follows, including members, connections, and web panels, which are listed in Tables A3–A5, respectively. It can be seen from Table A3 that in the initial design, the ratio of demand to capacity for columns labeled C2, C5, C9, and C10 and beams labeled B6 and B8 have exceeded 0.9, which is close to the ultimate, and the column labeled C7 is failed. At the same time, Table A5 shows that the web panels in the side columns from the third to the ninth floor are insufficient in shear. On the contrary, the beams labeled B1, B2 and B3 retain more margin in strength. After a round of adjustment for sections, in Iter1, the ratio of demand to capacity for all members is less than 0.9, and the remaining connections meet the performance requirements except for the connections labeled J19. However, unfortunately, despite the replacement with a larger section for beams or columns, there are still a large number of panels in side columns that are insufficient in shear.

A simple way would be to continue replacing with larger sections until all capacity requirements are met, but this study chose to strengthen these panels with welded double

plates. In Iter2, only the connection J9 and panel P9 have not yet got a satisfactory solution. Subsequently, the fixity-factor of this connection is amplified, and re-analysis is carried out to obtain the final scheme.

**Table A3.** Members and their most unfavorable ratio of demand to capacity in each iteration.

| No. | Initial | Iter1 | Iter2 | Iter3 |
|-----|---------|-------|-------|-------|
| C1 | HEB300<br>0.82 | HEB300<br>0.74 | HEB300<br>0.75 | HEB300<br>0.75 |
| C2 | HEB650<br>0.94 | HEB900<br>0.88 | HEB900<br>0.87 | HEB900<br>0.87 |
| C3 | HEB260<br>0.88 | HEB260<br>0.87 | HEB260<br>0.88 | HEB260<br>0.87 |
| C4 | HEB500<br>0.71 | HEB500<br>0.79 | HEB500<br>0.78 | HEB500<br>0.78 |
| C5 | HEB220<br>0.94 | HEB240<br>0.81 | HEB240<br>0.85 | HEB240<br>0.84 |
| C6 | HEB340<br>0.76 | HEB340<br>0.73 | HEB340<br>0.72 | HEB340<br>0.73 |
| C7 | HEB180<br>1.04 | HEB200<br>0.87 | HEB200<br>0.87 | HEB200<br>0.88 |
| C8 | HEB260<br>0.79 | HEB260<br>0.76 | HEB260<br>0.74 | HEB260<br>0.74 |
| C9 | HEB160<br>0.94 | HEB180<br>0.74 | HEB180<br>0.79 | HEB180<br>0.79 |
| C10 | HEB160<br>0.95 | HEB180<br>0.74 | HEB180<br>0.73 | HEB180<br>0.73 |
| B1 | IPE500<br>0.40 | IPE360<br>0.58 | IPE360<br>0.57 | IPE360<br>0.57 |
| B2 | IPE450<br>0.54 | IPE400<br>0.60 | IPE400<br>0.60 | IPE400<br>0.60 |
| B3 | IPE450<br>0.66 | IPE400<br>0.73 | IPE400<br>0.72 | IPE400<br>0.72 |
| B4 | IPE400<br>0.76 | IPE400<br>0.76 | IPE400<br>0.74 | IPE400<br>0.75 |
| B5 | IPE400<br>0.74 | IPE400<br>0.71 | IPE400<br>0.69 | IPE400<br>0.70 |
| B6 | IPE360<br>0.91 | IPE400<br>0.73 | IPE400<br>0.71 | IPE400<br>0.71 |
| B7 | IPE360<br>0.83 | IPE360<br>0.79 | IPE360<br>0.78 | IPE360<br>0.78 |
| B8 | IPE330<br>0.92 | IPE360<br>0.72 | IPE360<br>0.71 | IPE360<br>0.71 |
| B9 | IPE330<br>0.79 | IPE330<br>0.77 | IPE360<br>0.60 | IPE360<br>0.60 |
| B10 | IPE270<br>0.73 | IPE270<br>0.72 | IPE270<br>0.71 | IPE270<br>0.71 |

**Table A4.** Required and 'actual' connection properties in each iteration.

| Phase | No. | $r_{req}$ | $m_{req}$ | $r_{act}$ | $m_{act}$ | Check |
|-------|-----|-----------|-----------|-----------|-----------|-------|
| Initial | J1 | 0.70 | 0.31 | 0.72 | 0.67 | OK |
| | J2 | 0.70 | 0.40 | 0.71 | 0.74 | OK |
| | J3 | 0.75 | 0.47 | 0.77 | 0.79 | OK |
| | J4 | 0.70 | 0.54 | 0.71 | 0.69 | OK |
| | J5 | 0.75 | 0.50 | 0.76 | 0.65 | OK |
| | J6 | 0.75 | 0.66 | 0.77 | 0.78 | OK |
| | J7 | 0.75 | 0.60 | 0.76 | 0.72 | OK |
| | J8 | 0.75 | 0.76 | 0.77 | 0.85 | OK |
| | J9 | 0.75 | 0.58 | 0.77 | 0.61 | OK |
| | J10 | 0.75 | 0.74 | 0.77 | 0.77 | OK |
| | J11 | 0.80 | 0.68 | 0.82 | 0.76 | OK |
| | J12 | 0.80 | 0.91 | 0.81 | 1.05 | OK |
| | J13 | 0.80 | 0.56 | 0.82 | 0.60 | OK |
| | J14 | 0.80 | 0.83 | 0.81 | 0.86 | OK |
| | J15 | 0.80 | 0.63 | 0.82 | 0.65 | OK |
| | J16 | 0.80 | 0.92 | 0.81 | 0.94 | OK |
| | J17 | 0.80 | 0.52 | Null | Null | NO |
| | J18 | 0.80 | 0.79 | Null | Null | NO |
| | J19 | 0.85 | 0.41 | 0.87 | 0.80 | OK |
| | J20 | 0.85 | 0.70 | 0.87 | 0.80 | OK |

**Table A4.** *Cont.*

| Phase | No. | $r_{req}$ | $m_{req}$ | $r_{act}$ | $m_{act}$ | Check |
|-------|-----|-----------|-----------|-----------|-----------|-------|
| Iter1 | J1  | 0.80 | 0.51 | 0.81 | 0.80 | OK |
|       | J2  | 0.75 | 0.58 | 0.76 | 0.78 | OK |
|       | J3  | 0.75 | 0.48 | 0.74 | 0.78 | OK |
|       | J4  | 0.75 | 0.60 | 0.77 | 0.78 | OK |
|       | J5  | 0.75 | 0.57 | 0.76 | 0.72 | OK |
|       | J6  | 0.75 | 0.73 | 0.77 | 0.85 | OK |
|       | J7  | 0.75 | 0.60 | 0.76 | 0.72 | OK |
|       | J8  | 0.75 | 0.76 | 0.77 | 0.85 | OK |
|       | J9  | 0.75 | 0.58 | 0.77 | 0.69 | OK |
|       | J10 | 0.75 | 0.71 | 0.77 | 0.77 | OK |
|       | J11 | 0.75 | 0.56 | 0.77 | 0.69 | OK |
|       | J12 | 0.75 | 0.73 | 0.77 | 0.77 | OK |
|       | J13 | 0.80 | 0.58 | 0.82 | 0.69 | OK |
|       | J14 | 0.80 | 0.79 | 0.81 | 0.80 | OK |
|       | J15 | 0.80 | 0.52 | 0.82 | 0.69 | OK |
|       | J16 | 0.80 | 0.72 | 0.81 | 0.80 | OK |
|       | J17 | 0.80 | 0.54 | 0.82 | 0.65 | OK |
|       | J18 | 0.80 | 0.77 | Null | Null | NO |
|       | J19 | 0.85 | 0.46 | 0.84 | 0.80 | OK |
|       | J20 | 0.85 | 0.69 | 0.84 | 0.80 | OK |
| Iter2 | J1  | 0.80 | 0.51 | 0.81 | 0.80 | OK |
|       | J2  | 0.75 | 0.57 | 0.76 | 0.78 | OK |
|       | J3  | 0.75 | 0.48 | 0.74 | 0.78 | OK |
|       | J4  | 0.75 | 0.60 | 0.77 | 0.78 | OK |
|       | J5  | 0.75 | 0.65 | 0.77 | 0.73 | OK |
|       | J6  | 0.75 | 0.72 | 0.76 | 0.78 | OK |
|       | J7  | 0.75 | 0.69 | 0.77 | 0.73 | OK |
|       | J8  | 0.75 | 0.74 | 0.76 | 0.78 | OK |
|       | J9  | 0.75 | 0.67 | Null | Null | NO |
|       | J10 | 0.75 | 0.69 | 0.77 | 0.77 | OK |
|       | J11 | 0.75 | 0.64 | 0.77 | 0.65 | OK |
|       | J12 | 0.75 | 0.71 | 0.77 | 0.77 | OK |
|       | J13 | 0.80 | 0.58 | 0.82 | 0.62 | OK |
|       | J14 | 0.80 | 0.78 | 0.81 | 0.80 | OK |
|       | J15 | 0.80 | 0.51 | 0.82 | 0.62 | OK |
|       | J16 | 0.80 | 0.71 | 0.81 | 0.80 | OK |
|       | J17 | 0.80 | 0.48 | 0.82 | 0.65 | OK |
|       | J18 | 0.80 | 0.60 | 0.82 | 0.64 | OK |
|       | J19 | 0.85 | 0.46 | 0.84 | 0.80 | OK |
|       | J20 | 0.85 | 0.68 | 0.84 | 0.80 | OK |
| Iter3 | J1  | 0.80 | 0.51 | 0.81 | 0.80 | OK |
|       | J2  | 0.75 | 0.57 | 0.76 | 0.78 | OK |
|       | J3  | 0.75 | 0.48 | 0.74 | 0.78 | OK |
|       | J4  | 0.75 | 0.60 | 0.77 | 0.78 | OK |
|       | J5  | 0.75 | 0.66 | 0.77 | 0.73 | OK |
|       | J6  | 0.75 | 0.72 | 0.76 | 0.78 | OK |
|       | J7  | 0.75 | 0.69 | 0.77 | 0.73 | OK |
|       | J8  | 0.75 | 0.75 | 0.76 | 0.78 | OK |
|       | J9  | 0.80 | 0.62 | 0.82 | 0.73 | OK |
|       | J10 | 0.75 | 0.70 | 0.77 | 0.77 | OK |
|       | J11 | 0.75 | 0.64 | 0.77 | 0.65 | OK |
|       | J12 | 0.75 | 0.71 | 0.77 | 0.77 | OK |
|       | J13 | 0.80 | 0.58 | 0.82 | 0.62 | OK |
|       | J14 | 0.80 | 0.78 | 0.81 | 0.80 | OK |
|       | J15 | 0.80 | 0.51 | 0.82 | 0.62 | OK |
|       | J16 | 0.80 | 0.71 | 0.81 | 0.80 | OK |
|       | J17 | 0.80 | 0.48 | 0.82 | 0.65 | OK |
|       | J18 | 0.80 | 0.60 | 0.82 | 0.64 | OK |
|       | J19 | 0.85 | 0.46 | 0.84 | 0.80 | OK |
|       | J20 | 0.85 | 0.68 | 0.84 | 0.80 | OK |

**Table A5.** Required web panel in each iteration.

| Phase | No. | $M_{rd,req}$ | $M_{rd,act}$ | Continuity | Double | Check |
|---|---|---|---|---|---|---|
| Initial | P1 | 186.29 | 289.10 | 0 | 0 | OK |
| | P2 | 280.85 | 743.06 | 0 | 0 | OK |
| | P3 | 218.27 | 260.19 | 0 | 0 | OK |
| | P4 | 298.92 | 668.75 | 0 | 0 | OK |
| | P5 | 226.36 | Null | Null | Null | NO |
| | P6 | 359.40 | 495.55 | 0 | 0 | OK |
| | P7 | 208.37 | Null | Null | Null | NO |
| | P8 | 297.59 | 440.49 | 0 | 0 | OK |
| | P9 | 198.79 | Null | Null | Null | NO |
| | P10 | 262.66 | 274.46 | 0 | 0 | OK |
| | P11 | 182.91 | Null | Null | Null | NO |
| | P12 | 219.94 | 247.02 | 0 | 0 | OK |
| | P13 | 150.27 | Null | Null | Null | NO |
| | P14 | 170.40 | Null | Null | Null | NO |
| | P15 | 134.84 | Null | Null | Null | NO |
| | P16 | 114.40 | 149.70 | 0 | 0 | OK |
| | P17 | 110.52 | Null | Null | Null | NO |
| | P18 | 49.29 | Null | Null | Null | NO |
| | P19 | 53.28 | 58.16 | 0 | 0 | OK |
| | P20 | 16.65 | 58.16 | 0 | 0 | OK |
| Iter1 | P1 | 147.03 | 434.89 | 1 | 0 | OK |
| | P2 | 126.93 | 828.55 | 0 | 0 | OK |
| | P3 | 173.74 | 231.28 | 0 | 0 | OK |
| | P4 | 240.29 | 1319.40 | 1 | 0 | OK |
| | P5 | 198.36 | Null | Null | Null | NO |
| | P6 | 279.99 | 440.49 | 0 | 0 | OK |
| | P7 | 209.76 | Null | Null | Null | NO |
| | P8 | 300.47 | 440.49 | 0 | 0 | OK |
| | P9 | 202.23 | Null | Null | Null | NO |
| | P10 | 243.87 | 274.46 | 0 | 0 | OK |
| | P11 | 194.74 | Null | Null | Null | NO |
| | P12 | 233.22 | 274.46 | 0 | 0 | OK |
| | P13 | 156.48 | 190.01 | 1 | 0 | OK |
| | P14 | 148.18 | 163.31 | 0 | 0 | OK |
| | P15 | 140.44 | 190.01 | 1 | 0 | OK |
| | P16 | 111.49 | 163.31 | 0 | 0 | OK |
| | P17 | 117.03 | Null | Null | Null | NO |
| | P18 | 47.66 | Null | Null | Null | NO |
| | P19 | 59.64 | 66.90 | 0 | 0 | OK |
| | P20 | 17.02 | 66.90 | 0 | 0 | OK |
| Iter2 | P1 | 146.16 | 434.89 | 1 | 0 | OK |
| | P2 | 125.53 | 828.55 | 0 | 0 | OK |
| | P3 | 172.61 | 231.28 | 0 | 0 | OK |
| | P4 | 236.11 | 1319.40 | 1 | 0 | OK |
| | P5 | 225.73 | 267.92 | 0 | 1 | OK |
| | P6 | 269.17 | 440.49 | 0 | 0 | OK |
| | P7 | 237.19 | 267.92 | 0 | 1 | OK |
| | P8 | 285.71 | 440.49 | 0 | 0 | OK |
| | P9 | 229.46 | Null | Null | Null | NO |
| | P10 | 228.55 | 274.46 | 0 | 0 | OK |
| | P11 | 220.40 | 242.47 | 0 | 1 | OK |
| | P12 | 219.20 | 274.46 | 0 | 0 | OK |
| | P13 | 155.29 | 190.01 | 1 | 0 | OK |
| | P14 | 144.60 | 163.31 | 0 | 0 | OK |
| | P15 | 139.08 | 190.01 | 1 | 0 | OK |
| | P16 | 108.86 | 163.31 | 0 | 0 | OK |
| | P17 | 129.32 | 137.47 | 0 | 1 | OK |
| | P18 | 45.36 | 89.20 | 0 | 0 | OK |
| | P19 | 59.26 | 66.90 | 0 | 0 | OK |
| | P20 | 14.81 | 66.90 | 0 | 0 | OK |

**Table A5.** *Cont.*

| Phase | No. | $M_{rd,req}$ | $M_{rd,act}$ | Continuity | Double | Check |
|-------|-----|------|------|------------|--------|-------|
| Iter3 | P1 | 146.21 | 434.89 | 1 | 0 | OK |
| | P2 | 125.60 | 828.55 | 0 | 0 | OK |
| | P3 | 172.70 | 231.28 | 0 | 0 | OK |
| | P4 | 236.31 | 1319.40 | 1 | 0 | OK |
| | P5 | 226.33 | 267.92 | 0 | 1 | OK |
| | P6 | 269.79 | 440.49 | 0 | 0 | OK |
| | P7 | 237.18 | 267.92 | 0 | 1 | OK |
| | P8 | 287.59 | 440.49 | 0 | 0 | OK |
| | P9 | 213.45 | 297.32 | 1 | 0 | OK |
| | P10 | 232.30 | 274.46 | 0 | 0 | OK |
| | P11 | 220.27 | 242.47 | 0 | 1 | OK |
| | P12 | 221.81 | 274.46 | 0 | 0 | OK |
| | P13 | 155.68 | 190.01 | 1 | 0 | OK |
| | P14 | 144.77 | 163.31 | 0 | 0 | OK |
| | P15 | 139.13 | 190.01 | 1 | 0 | OK |
| | P16 | 108.92 | 163.31 | 0 | 0 | OK |
| | P17 | 129.39 | 137.47 | 0 | 1 | OK |
| | P18 | 45.32 | 89.20 | 0 | 0 | OK |
| | P19 | 59.28 | 66.90 | 0 | 0 | OK |
| | P20 | 14.79 | 66.90 | 0 | 0 | OK |

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
