# Peer review of "A Design Method for Semi-Rigid Steel Frame via Pre-Established Performance-Based Connection Database"

_buildings, doi:10.3390/buildings12101634_

Round 1
Reviewer 1 Report
Dear authors,
A design method for semi-rigid steel frame via pre-established performance-based connection database is a very interesting manuscript. In order for it to be accepted for publication, I would recommend the following:
- extensive English check and proof (nouns instead of adjectives or verbs, e.g., capable/capability line 33, adjective instead of adverb, e.g., significantly/significant, line 33, tenses, line 47, 197, 298, unnecessary emphasis, line 47, readability/understanding of the text – lines 35, 42-43, 56-59, colloquial, line 82, 127, 223, 259, 694, 699 etc., spelling, e.g., form/from, line 336 and so on);
- line 163 – assign the corresponding reference;
- the study is well integrated within the literature, but I suggest that the large part of the theoretic approach from chapters to be split, to have a good picture of the research part;
- include information on the software + version used for data obtained, if the case;
- assign the figures and the tables (include the appropriate source references where the case).
Reviewer 2 Report
(1) The paper focuses on semi-rigid beam to column join connections in steel frame buildings. In particular, the study provides a design method based on the development and the use of a performance-based connection database and characterization/classification criteria. The study relies on and extends a literature study carried out by some of the Authors [34]. In particular, fixity factor and moment capacity coefficient are considered as reference parameters for developing the database and implementing the methodology. The sensitivity of the structural response is firstly assessed as a function of the variation of the reference parameters, in order to define the parameter range for the methodology implementation. The design methodology is finally applied with regard to two literature frame buildings, showing that its implementation can be efficient and satisfactory.
(2) The study might potentially contribute to the literature and practice, especially with regard to technical and professional applications, even though the scientific contribution is relatively reduced. In particular, the developed methodology could be useful and efficient for designing steel frame buildings (beam to column joints). However, the manuscript should be reconsidered after major revisions, as reported in the following.
(3) (a) One of the most critical issues of the paper is associated with a general unclearness and inefficiency of the text to deliver the contents; several syntax, grammar, and typo issues affect the paper, and the text flow is not simply to follow; the paper reports heavy technical and applicative data/information that make harder the understanding. (b) The scientifical contribution of the study should be clearly identified and discussed in the paper since the paper is expected to provide original scientific contributions; the original contribution should be clearly claimed with regard to past studies and previous studies carried out by the authors [34]. (c) The results should be reported and discussed in a more efficient manner, e.g., by summarizing/synthesizing the single outcomes and by highlighting their significance/impact. (d) Please, also refer to the reviewed manuscript report for detailed review comments and recommendations.

Reviewer 3 Report
The reviewed manuscript deals with the problem of the design of steel frames with semi-rigid connections. The presented approach is original (provides a new look at the problem) and interesting. The article is well written and does not need linguistic corrections. Please consider the following comments.
Main doubts and remarks:
From the reviewer's point of view, the part with the examples is too large. Especially the example with multi-storey frame contains a lot of Tables which while reading are not easy to follow. The three-storey frame example would be more convenient.
In the conlcusions, you wrote "The structural design is transformed into a search for the proper matching of performance" - what serching method was used?
It is not clear how the internal forces and displacements were found - could provide relevant information about that?
While presenting the examples it would be important to add information about the step of your approach with respect to Figure 3. It will significantly increase the readability of this part of the examples. Please consider this remark very important.
Please specified what kind of design criteria you used in your examples.
It is extremely important for the reader to present the comparison of the results from your method with results from other available engineering software.
Additional remarks and reviewer's suggestions:
line 125 - the additional information about how the initial rotational stiffness should be determined would be welcomed
Figure 1: please mention in the text what kind of global analysis should be used for 3. Structural analysis
Figure 3 and Figure 1 are very simmilar. The reviewer recommends to highlight the diffrences in Figure 3 with respect to Figure 1.
Derivation of the formula 6 would be welcomed.
Editorial errors:
line 42 - please use "a" instead "A"
line 163: references are missing in pdf file ...
Round 2
Reviewer 2 Report
The referee is satisfied with the revised version of the manuscript: the quality of the paper is significantly enhanced. The paper is now suitable for publication after a final grammar/spell check is carried out.
Author Response
Dear Reviewer,
Thank you for your letter dataed 20 Sep. 2022. We were pleased to know that our work was rated as potentially accecptable for publication in Journal, subject to adequate check. Thank again the time and effort that you have put into reviewing the previous version of the manuscript.
After careful examination, we have made the following updates base on the previous version, please check by using the “track changes” mode in MS Word.
The numbers of page and line are corresponding to MS “track changes” mode. Before checking, please select the “All markup”.
1. Page 1, line 12, remove “the”.
2. Page 1, line 13, remove “the”.
3. Page 1, line 14, add “s” after “method”.
4. Page 1, line 16, remove “s” after “element” and “connection”.
5. Page 1, line 17, add “s” after “frame”.
6. Page 1, line 19, remove “s” after “frame”.
7. Page 1, line 20, add “ing of performance requiredments” after “match”; remove “performance requirements” after “ connections”.
8. Page 1, line 22, move “only slightly” in front of “affected”.
9. Page 1, line 24, add “the” before “connection”; remove “s” after “connection”.
10. Page 1, line 26, add “In addition,”; remove “Compared with tuning……of connections,”.
11. Page 1, line 27, replace “their” with “the”; add “requirements of the connection” after “performance”.
12. Page 1, line 28, add “compared to tuning its geometry” after “design variables”; add “it” before “provides”.
13. Page 2, The space for symbols table has been adjusted.
14. Page 2, line 145, add “the” before “semi-rigid steel frame”.
15. Page 7, line 224, add “the” before “fixity-factor”.
16. Page 12, line 365, add “(a) (b)”.
17. Page 12, line 366, remove “The”, replace “c” with “C”.
18. Page 15, line 418, remove “The”, replace “i” with “I”.
19. Page 16, line 422, remove “The”, replace “i” with “I”.
20. Page 21, line 614, remove “The”, replace “l” with “L”.
21. Page 25, line 690, 691, replace “Required” with “Actual”.
22. Page 35, line 858, add “, 52278181”.
Reviewer 3 Report
I would like to kindly thank the Authors for their answers and manuscript improvements. The improvements had been done.
Nevertheless, the reviewer still has doubts about the transparency of the article. In my opinion, there is no logical and coherent development of the very interesting idea presented in Figure 2.
I would like to leave the decision of the acceptance of the article to the editor.
Reviewer 4 Report
Unfortunately, there is no response letter provided by the authors and the modifications are not visible; hence, no further review can be made on the revised manuscript.
Round 3
Reviewer 4 Report
The manuscript may be accepted after minor revision including the language editing.